# Community Agricultural Reservoir Construction and Water Supply Network Design in Ubon Ratchathani, Thailand, Using Adjusted Variable Neighborhood Strategy Adaptive Search

**Rerkchai Srivoramasa [1], Natthapong Nanthasamroeng [2,*], Rapeepan Pitakaso [3], Thanatkij Srichok [3], Surajet Khonjun [3], Worapot Sirirak [4] and Chalermchat Theeraviriya [5]**

1 Department of Civil Engineering, Faculty of Engineering, Ubon Ratchathani University, Ubon Ratchathani 34190, Thailand
2 Artificial Intelligence Optimization SMART Laboratory, Department of Engineering Technology, Faculty of Industrial Technology, Ubon Ratchathani Rajabhat University, Ubon Ratchathani 34000, Thailand
3 Artificial Intelligence Optimization SMART Laboratory, Department of Industrial Engineering, Faculty of Engineering, Ubon Ratchathani University, Ubon Ratchathani 34190, Thailand
4 Department of Industrial Engineering, Faculty of Engineering, Rajamangala University of Technology Lanna, Chiang Rai 50300, Thailand
5 Department of Industrial Engineering, Faculty of Engineering, Nakhon Phanom University, Nakhon Phanom 48000, Thailand
* Correspondence: natthapong.n@ubru.ac.th

**Abstract:** Agricultural sectors all over the world are facing water deficiencies as a result of various factors. Countries in the Greater Mekong Subregion (GMS) in particular depend on the production of agricultural products; thus, drought has become a critical problem in such countries. The average water level in the lower part of the Mekong River has been decreasing dramatically, resulting in the wider agricultural area of the Mekong watershed facing a lack of water for production. The construction of community reservoirs and associated water supply networks represents a strategy that can be used to address drought problems in the GMS. This study aims to solve the agricultural community reservoir establishment and water supply network design (CR–WSND) problem in Khong Chiam, Ubon Ratchathani, Thailand—a city located in the Mekong Basin. The CR–WSND model is formulated using mixed-integer programming (MIP) in order to minimize the cost of reservoir construction and water irrigation. An adjusted variable neighborhood strategy adaptive search (A-VaNSAS) is applied to a real-world scenario involving 218 nodes, and its performance is compared with that of the original variable neighborhood strategy adaptive search (VaNSAS), differential evolution (DE), and genetic algorithm (GA) approaches. An improved box selection formula and newly designed improvement black boxes are added to enhance the quality beyond the original VaNSAS. The results reveal that the quality of the solution from A-VaNSAS is significantly better than those of GA, DE, and VaNSAS (by 6.27%, 9.70%, and 9.65%, respectively); thus, A-VaNSAS can be used to design a community reservoir and water supply network effectively.

**Keywords:** water reservoir; location–allocation sizing problem; genetic algorithm; differential evolution; variable neighborhood strategy adaptive search

## 1. Introduction

A drought is defined as a period of unusually dry weather lasting long enough to produce major issues such as agricultural damage and/or water shortages. Droughts are brought on by a lack of precipitation over a long period of time caused by a variety of factors, including climate change, ocean temperatures, and changes in the local topography. Thailand has been suffering from its worst drought in at least four decades, especially in the northeast. Ubon Ratchathani is a province located in the northeast of Thailand as well as being located in the Mekong Basin, as shown in Figure 1.

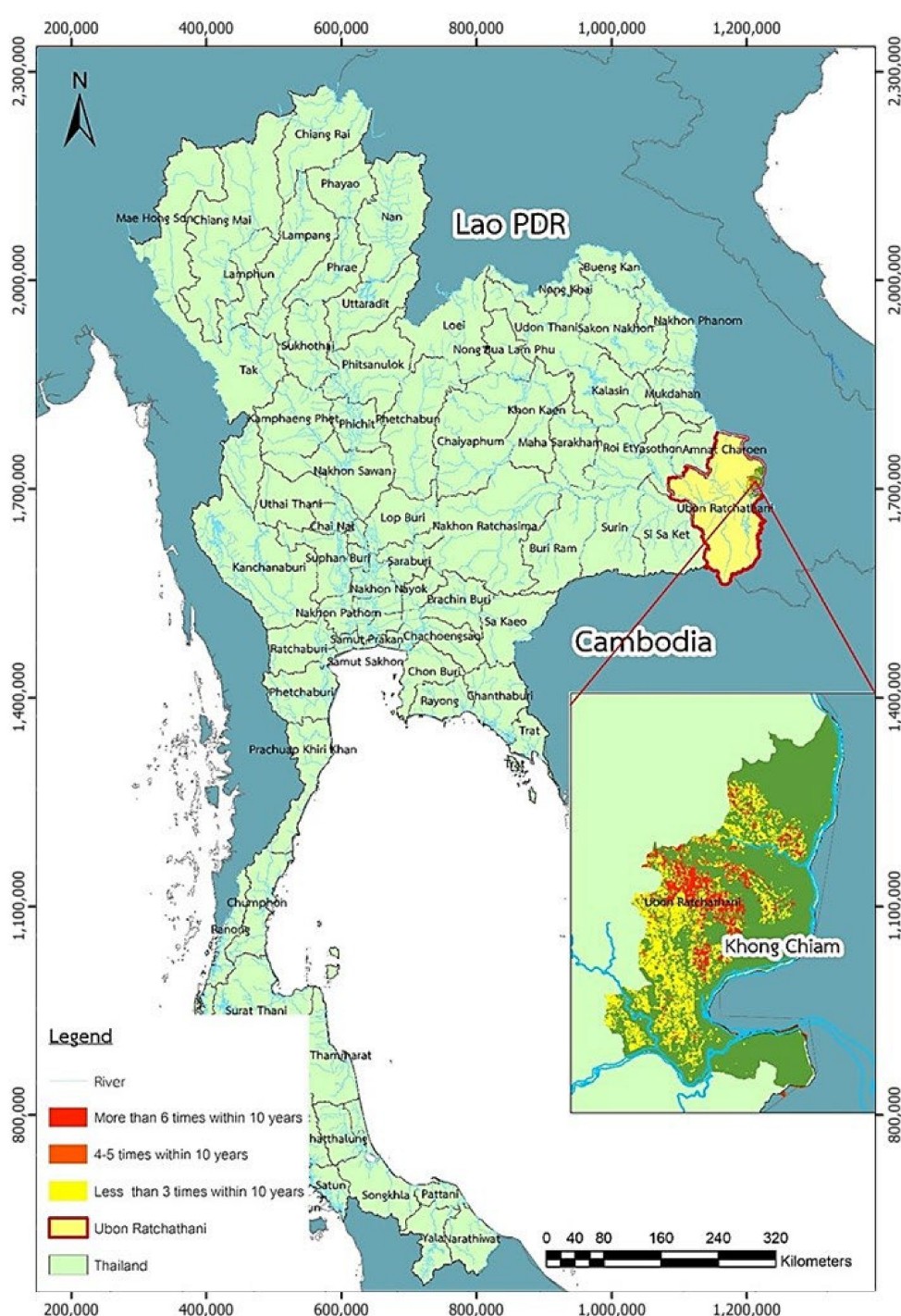

**Figure 1.** Map of Thailand showing the study location.

There are a great number of tributary rivers in the Mekong Basin, including the Ruak, Kok, Ing, Mun, and Chi rivers. In the past five years, in northeastern Thailand and the Lao PDR, the water levels of the Mekong rivers have been significantly below average. Levels at mainline measurement sites in Chiang Saen, Chiang Khan, Luang Prabang, Vientiane, Nong Khai, Nakhon Phanom, Mudaharn, and Ubon Ratchathani are all lower than they were previously. All of the mainstream water levels north of Stung Treng are much lower than average for the time of year and have been predicted to continue to fall. Similarly, river levels in southwestern China have been at their lowest in 50 years with water flowing at about half of what is considered average for February. The drought conditions in northern

Thailand and Lao PDR have resulted in low water levels on the main Mekong as part of a larger regional drought upstream in China's Yunnan Province.

The lower water level in the Mekong enhances water flow from the Mekong's tributary rivers. The result of this is that the large reservoirs along the rivers must provide water to the rivers in order to maintain water usage, especially throughout the agricultural production process, which is the primary occupation of people living in Ubon Ratchathani and other provinces in northeastern Thailand. The large reservoirs constructed in the northeast of Thailand comprise around 11,462 million cubic meters, while the water use requirement in the area is around 10,815 million cubic meters. Thus, the northeastern part of Thailand has not been considered to be likely to experience water supply problems. However, according to the statistics of the drought reported, in the northeastern part of Thailand, especially Ubon Ratchathani, more water is needed for agricultural production—around 2496 million cubic meters. It has been proven that the amount of water available during the year is theoretically sufficient; however, due to the lower level of the Mekong, water from its tributary rivers tends to flow down into the Mekong and finally down to the South China Sea through Vietnam as a result of the drought in this area [1].

Ubon Ratchathani is a province in northeast Thailand located in the Mun Basin. The Mun River is one of the three main tributary rivers of the Mekong. Its watershed area covers more than 69,700 square kilometers. The Land Development Department, Ministry of Agriculture and Cooperatives, Royal Thai Government, has reported that the Mun River Basin contains large and small reservoirs with a water retention capacity of around 3979 million cubic meters, which is not enough to meet the water use demand in the area [2]. The water required in this area equates to around 3061 million cubic meters, which is lower than the full capacity of all the reservoirs established in the area; nevertheless, the Department of National Parks, Ministry of Natural Resources and Environment, Royal Thai Government [1], has reported that in this area, the lack of water supply is still the main problem for agriculture, and the extra water needed in this area equates to more than 1188 million cubic meters annually. This is because the lower water level of the Mekong causes the Mun River to flow down faster to the Mekong, making drought occur in the area.

The Thai government plans to invest THB 1408 million in constructing small community reservoirs (CRs) in the next few years in order to retain as much rainwater as possible, as well as using underground water to fulfill the water supply demand. The key aims of this research are: (1) to design community reservoirs for the target area; (2) to discover the correct number, size, and location of the CRs; and (3) to design a network for the located reservoirs in order to distribute water from the CRs to neighboring agricultural communities that do not have a CR.

Figure 2 shows a drought risk plot of Khong Chiam City. Khong Chiam is located in the Mun Basin and is the last city in Thailand before the water from the Mun River flows down to the Mekong. Khong Chiam has a vast drought risk area, around 1,217,600 m$^2$ [1], which must be managed before it has a significant effect on the agricultural production supply chain. Drought has been particularly harmful to the rice yield in the Mun River Basin, which has the lowest rice yield in Asia [3]. Future arid conditions in this region are expected to have a greater impact on the agricultural production supply chain [4]. Along the Mun River and between the provinces of Nakhon Ratchasima and Ubon Ratchathani, seven large dams have been constructed. Even though they can gather more than 3900 million cubic meters of water, the number of regions with a high danger of drought is expanding. This suggests that the water supply in this region must be revised in order to preserve the agricultural supply chain from damage caused by drought. In many regions of the world where hydrological variability is considerable, Hughes, D. and Mantel, S. [5] have demonstrated that even a small reservoir can boost water supply reliability. Thus, we expect that increasing the number of small reservoirs will have a positive effect on the downstream flow volumes and patterns of water in the examined area.

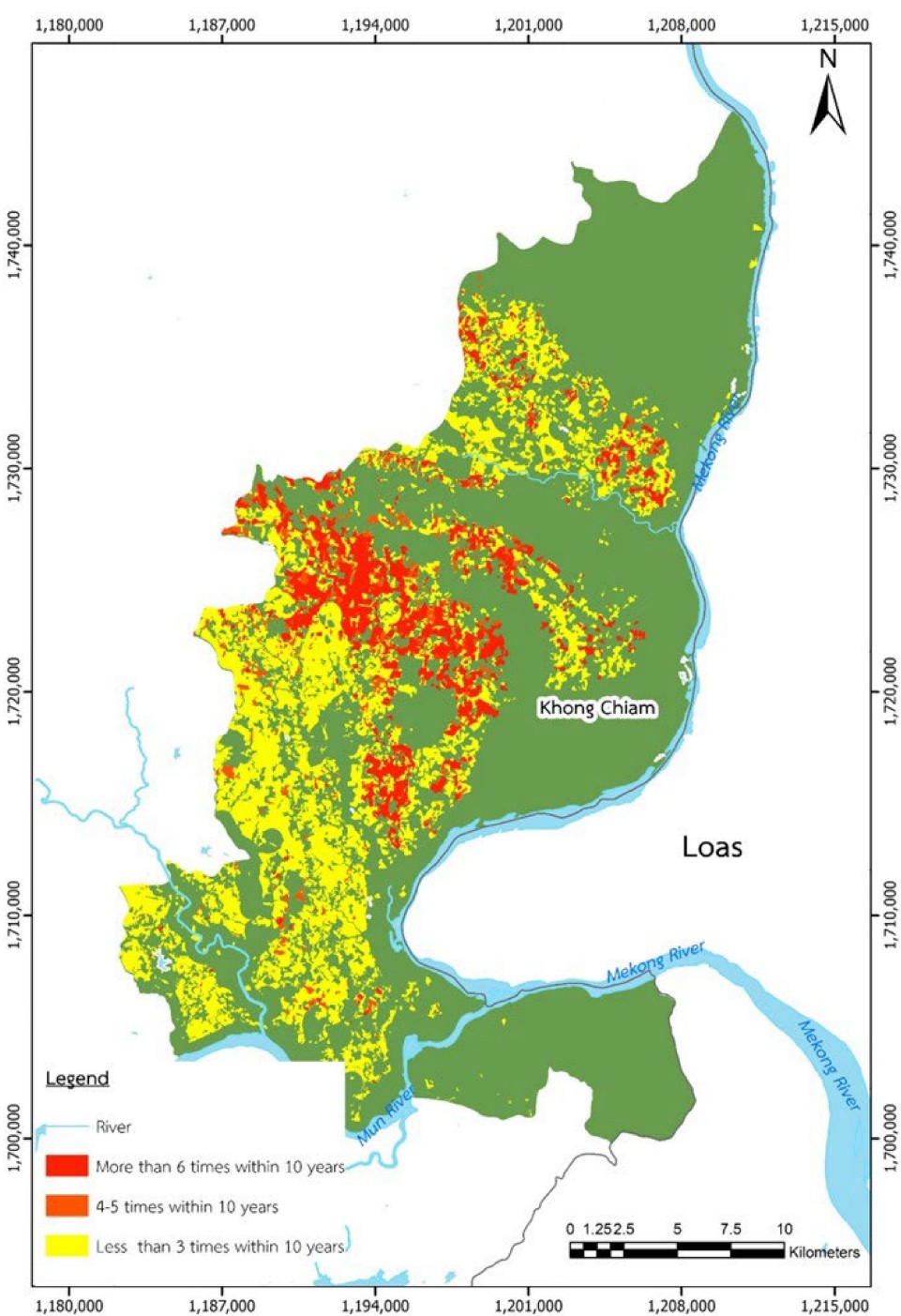

**Figure 2.** Khong Chiam City, Ubon Ratchathani, Thailand (case study).

The construction of small reservoirs in regions of the Mun River Basin with a high risk of drought may lessen the likelihood that these regions experience drought. We were unable to discover any research demonstrating the consequences of constructing small reservoirs in the Mun River Basin based on an examination of existing research on drought risk management in the Mun River Basin. Thus, our first research question is: "Can establishing small reservoirs in the Mun River Basin, particularly in the province of Ubon Ratchathani, reduce the likelihood of drought in this area and the impact of drought on agricultural goods production?"

As shown in Figure 2, the regions of Ubon Ratchathani with a high risk of drought are widespread. Establishing small reservoirs in a certain region might not alleviate the problem in all of the nearby regions with a high risk of drought. Do Guen Yoo et al. [6] presented a model for identifying the optimal pipe diameter to connect the reservoir region to the water demand points. They discovered that the optimal water supply pipe size has lower construction costs than the existing size while still ensuring adequate water supply. This research leads us to combine two difficult problems: (1) the problem of determining the optimal placement of small reservoirs and (2) the problem of constructing a pipe water supply network connected to the established small reservoirs in order to deliver water to water demand sites. It is anticipated that by solving these interrelated problems, the chance of drought may be reduced. This prompts us to expand our first research question to: "Can creating small reservoirs in the right locations and designing a pipe water supply network to connect the established reservoirs reduce the number of high-drought-risk areas in the province of Ubon Ratchathani?"

The combination of these two problems is referred to as the capacitated location–allocation problem and is a form of a logistics network design problem. In general, the location–allocation problem involves building a logistics network that can satisfy demand at all demand points, and the objective of a typical location–allocation problem is to minimize operating costs [7,8]. In this study, the model's objective function considers not only the lowest operating/construction costs but also the total area of the high-drought-risk locations. In addition, the number of established reservoirs and their total capacity are unknown in this research, yet Church, R. and ReVelle, C. [9] and Drezner, Z. [10] created models to decrease the maximum network distance when the number of established locations is known. Capacitated location–allocation problems are known to be NP-hard [11]; thus, many researchers have attempted to solve them using heuristic approaches such as a genetic algorithm (GA) [11,12], differential evolution algorithm (DE) [13,14], evolutionary simulated annealing (ESA) [15], and simulated annealing (SA) [16,17].

Pitakaso, R. et al. [7] recently introduced the variable neighborhood strategy adaptive search (VaNSAS) method to address the logistics design network for agricultural products in the northeast of Thailand. The authors demonstrated that VaNSAS can outperform SA and other heuristics, such as variable SWAP (VSWAP) and iterated local search (ILS) in terms of locating the optimal solution. They even demonstrated the effectiveness of VaNSAS in solving combinatorial optimization problems, but also revealed the algorithm's flaws; namely, it can only find a satisfactory solution during the last phase of the simulation. This is due to the absence of guidance from the current best solution when selecting an improvement technique in VaNSAS, which results in sub-optimal behavior of the intensification search. Thus, in this study, the formula used to select the improvement methods is modified to incorporate the guidance from the current best solution with the expectation that the modified version of VaNSAS—adjusted VaNSAS (A-VaNSAS)—will improve upon the solution quality of the original VaNSAS; this is the second research question of this study.

In the following section, a literature review and related work are described. In Section 3, the CR design is presented; then, in Section 4, the MIP mathematical model formulation for the reservoir's water supply network design is detailed. Finally, the proposed method, computational results, and conclusion are provided in Sections 3–5, respectively.

## 2. Literature Review and Related Works

In this section, a systematic review of previous articles related to the proposed problem is detailed. The adjusted variable neighborhood strategy adaptive search (VaNSAS) is used to select a location suitable for locating the community reservoirs and designing their water supply networks to solve the drought problem in Khong Chiam City, Ubon Ratchathani, Thailand. This city is located in the Mekong River Basin. The review of the previous research provided in this section is divided into three subsections: (1) the Mekong River Basin; (2) drought management; and (3) VaNSAS and metaheuristics.

### 2.1. Mekong River Basin

The Mekong River originates on the Tibetan Plateau and runs through China's Yunnan region, Myanmar, Lao People's Democratic Republic, Thailand, Cambodia, and Vietnam before emptying into the South China Sea. With a length of 4909 km, a drainage area of 795,000 km$^2$, and an average annual discharge of 14,500 m$^3$/s, the Mekong River is one of the world's longest and greatest rivers [18,19]. Plateaus, terraces, hills, valleys, and mountains dot the upper Mekong River and its environs, and the land-cover types shift from grassland to a mix of farmland, agroforestry, and shrubs from west to east. Myanmar, Thailand, Vietnam, Cambodia, and parts of India are all part of the Lower Mekong River Basin and its neighboring areas. It is densely forested and is characterized by a large amount of agriculture.

Over the last ten years, the water level of the Mekong River has dropped drastically. One explanation for this is that twelve hydroelectric dams have been erected (with a few more in the works) obstructing the flow and filling upstream, particularly upstream of China, with around 70% of the water obstructed and the flow shifted.

Furthermore, the availability of water in this region is particularly vulnerable to the effects of climate change [20]. Climate change may result in unanticipated water phenomena in the river. As a result, the flow is dry, particularly in downstream nations such as Thailand, Laos, and Vietnam, during the dry season. Drought is caused by a shortage of water, which promotes saline intrusion and prevents the accumulation of alluvium, resulting in nutrient poverty for plants and ecosystems [21].

The Mun River in Thailand is the Mekong River's greatest tributary, supplying roughly $20 \times 10^9$ m$^3$ of water to the Mekong River each year (Li et al., 2020). As a result, changes in the Mun River's water contribution can have a significant impact on the Mekong River's water resources in the middle and lower sections. The Mun River Basin (MRB) is a major agricultural zone both locally and worldwide, with agricultural land accounting for over 80% of the whole basin area. Due to water scarcity, rain-fed rice yields in the MRB are often lower than potential yields [3,22]. Changes in streamflow as a result of future climate change may have an impact on agricultural water demand in this area [23], increasing hydrological uncertainty and complicating local water resource management.

In this study, we aim to control the water supply for agricultural areas in the Mun River Basin at a local level. When the water supply from the major reservoirs and the main river is cut off, the community reservoirs are to be used to collect natural water in certain spots. Following this method, the drought issue may be alleviated to some degree.

### 2.2. Drought Management

Drought is a global issue that is becoming more urgent every year. Water scarcity was expected to have affected more than one billion people globally by 2020 [24]. Drought types can be classified into three categories: meteorological, hydrological, and agricultural [25]. Drought in agriculture mostly impacts agricultural production, which is important for food security, the economy, and stability [26]. Many papers have been written on drought management from both social science and engineering standpoints.

The majority of past research has focused on the impact of drought on society and the cooperation of parties involved in drought areas from a social science perspective. The construction of a large dam in the high reaches of a river may cause drought in the lower reaches of the river, thus creating conflict between villages. As a result, conflict management is required to deal with such a circumstance [27]. Other social science-based suggestions for reducing the effects of drought have been proposed, including: (1) drought risk management organizations, which make it easier for different water and land users, meteorologists, and watershed management professionals to communicate with one another and (2) communication and discussion between all stakeholders in a particular region about the present drought situation in order to determine whether regular water scarcity periods have occurred or whether the current situation is the result of persistent and severe drought conditions [28–31].

Other research has focused on the operation and construction of huge reservoirs from an engineering standpoint. A reservoir, in theory, can mitigate the effects of droughts. Since it affects hydrological systems and downstream socio-economic aspects as well as the environmental sustainability of local lakes and wetlands, the construction of a new dam may not be a comprehensive solution to preventing drought [32–34]. Drought— defined as periods of extreme scarcity of water that significantly influence human activities or environmental needs—can be exacerbated (or induced) by the construction of large reservoirs [35,36]. This can lead to a disparity in water availability and, as a result, societal pressure to build more reservoirs, further exacerbating the problem. Both large, publicly controlled reservoirs and smaller, privately held reservoirs can play a role in such a process. Understanding the hydrological impact of a dense network of (small) reservoirs (DNR) is critical from both a socio-economic and water management standpoint.

In many parts of the world, small reservoirs and accompanying water supply networks are the preferred solution to the drought problem, such as in Australia [37], Northeast Brazil [38], Ethiopia [39], Ghana [40], India [41], South Africa [5], South Brazil [42], and Thailand [43,44]. Small reservoirs are unlikely to have a significant impact on a hydrological system as their maximum storage capacity is minimal. In fact, the cumulative effect of tiny reservoirs in resolving the drought problem may be greater than the influence of a single large reservoir.

In this study, small reservoirs are employed for placement in high-drought-risk locations in order to mitigate the potential impacts of drought. The water supply network of minor reservoirs is to be developed as the water supply network of community reservoirs after they are built. The reservoirs employed in this study are considered to be of three different sizes, all of which are adapted from the clay pond designed by Thailand's Ministry of Agriculture and Cooperatives Land Development Department. In the decision-making field, the proposed problem is viewed as a location–sizing problem combined with a location–allocation problem. The location–allocation problem, often known as the facility location problem [45,46], is NP-hard. As a result, an efficient method is required to handle the complex problem of facility localization. A review of metaheuristics methods, particularly the variable neighborhood strategy adaptive search (VaNSAS), is undertaken in the following subsection.

### 2.3. VaNSAS and Metaheuristics

A metaheuristic is a higher-level technique or heuristic used to locate, produce, or select a heuristic that can provide a good solution to an optimization issue, especially when there is incomplete or defective information or limited computational capability [47,48]. Metaheuristics can solve optimization problems faster than exact methods such as simplex, branch and bound, branch and cut, etc. The downside of metaheuristics is that they do not promise that the solution will always be the best, in contrast to exact approaches, which guarantee that the best solution will always be found. Exact methods cannot always be used to tackle real-world problems, which may be too large and complex. As a result, metaheuristics are becoming increasingly popular as they can deliver high-quality solutions in a short amount of time.

The genetic algorithm (GA), particle swarm optimization (PSO), differential evolution algorithm (DE), simulated annealing (SA), and tabu search (TS) are examples of well-known metaheuristics. Such methods have been utilized to handle a variety of issues, including vehicle routing problems, traveling salesman problems, lot sizing problems, and facility location problems. The genetic algorithm (GA), particle swarm optimization (PSO), flower pollination method, simulated annealing (SA), and improved harmony search algorithms are heuristic approaches that have been utilized in past research to solve location and sizing problems [49–53]. The adjusted variable neighborhood strategies adaptive search (A-VaNSAS) is used to solve the proposed problem in this study.

Pitakaso et al. [7] initially proposed VaNSAS to solve the location routing problem. VaNSAS includes four steps: (1) generate an initial set of tracks; (2) run the track touring

procedure; (3) update the heuristics information; and (4) repeat steps 2 and 3 until the termination condition is reached. The basic idea behind VaNSAS is that several types of heuristics are used to improve the quality of the present solution. Metaheuristics, basic heuristics, and the well-known local search technique are among the heuristics utilized in VaNSAS, and three to four heuristics are usually constructed. In black box optimization, the track chooses the heuristic independently. With varied probability, an appropriate improvement process (IP) is chosen. The likelihood of selecting an IP is changed repeatedly based on the average solution quality of the tracks that have previously used that IP. VaNSAS has been utilized to handle a variety of problems, including the location routing problem [54], assembly line balance problem [55], and scheduling and routing problem [56].

*2.4. Most Recent Research in Reservoir Construction and Water Supply Network Design*

Existing study proposals about reservoir construction and water supply network design are mostly concerned with the environmental impacts that have occurred or may occur when a reservoir was or is constructed. Residents' perspectives on the effects of the Metolong Dam and Reservoir were studied by Sekamane et al. [57]. The qualitative methodology utilized document analysis, field notes, and semi-structured interviews. The findings indicated that locals saw the overall social, economic, and environmental effects of the dam and reservoir as mixed. Environmentally, the area was impacted by noise and air pollution, soil erosion, and habitat loss during construction, despite efforts to protect endangered species. Ahmed, M., Cho, G. and Choi, K. [58] assessed the performance of 400 main agricultural reservoirs in South Korea as a function of climate change from 1973 to 2017, taking into account the components of reservoir water balance, including watershed runoff, irrigation water demand, and evaporation loss. Using the Territorial Life Cycle Assessment (T-LCA) methodology, Rogy et al. [59] sought to evaluate the conditions wherein hydraulic projects may be viewed as an environmentally efficient choice for securing the water supply of agricultural areas. Zhang et al. [60] proposed a new integrated modeling approach to estimate the agriculture water supply risk in the Baojixia Irrigation Area (BIA) of northwest China. Their results showed that this integrated methodology is a complete, modern, and efficient instrument for assessing farm water supply risk. A weak correlation between BIA precipitation and upstream runoff might create unpredictability in natural and irrigated agriculture water availability.

Existing research has also focused on predicting or forecasting the amount of water in a dam or large reservoir in order to plan for reducing or maintaining the volume of water in the large reservoir. Using AI approaches, artificial neural networks (ANN), support vector regression (SVR), and long short-term memory (LSTM), de Araújo et al. [61] estimated the reservoir volumes of two reservoirs (the Ladik and Yedikir Reservoirs). Various parameters' influence on the performance of the models were studied. Using artificial neural networks and data from 141 monitored dams, Özdoğan-Sarıkoç et al. [62] determined that the implementation of a water management system dramatically increased the siltation ratio. Since the commencement of this management, water withdrawal has increased, resulting in decreased reservoir levels at the beginning of the rainy season. This decreased the likelihood of overflowing—the primary mechanism of sediment discharge—resulting in excessive siltation.

The following contributions are made in this study: (1) A mathematical formulation of the CR–WSND problem regarding an agricultural water reservoirs is introduced; (2) A-VaNSAS is adjusted for application to the proposed problem; and (3) a real case study facing drought problems is effectively solved. A mathematical model is presented in the next section, demonstrating the proposed problem.

## 3. Research Methodology

This study has two primary goals: (1) to present a model to reduce a high-drought-risk area by utilizing a community reservoir water supply network and (2) to present an efficient

approach for solving the suggested model. Executing these two objectives requires two major parts of the research methodology: model building and model testing.

### 3.1. Model-Building Phase

The model-building phase presented in this paper is composed of two parts: (1) the formulation of a mathematical model to represent the proposed model in order to solve the high drought risk area problem and (2) the adjusted variable neighborhood strategy adaptive search (A-VaNSAS) in order to solve the proposed mathematical model.

First, the model representing the considered problem was mathematically constructed. A mathematical model is a model that represents the problem statement and the requirements or constraints of the designed model. The capacitated location–allocation problem provided by Chandra, S. et al. [63] and Demir, I. et al. [64] was modified to be more suitable and applicable to the community reservoirs and the associated water supply network given their individual limitations. The original purpose of both papers mentioned above was to tackle the capacitated facility location–allocation problem for wastewater treatment in an industrial cluster and the multi-objective capacitated multiple allocation hub location problem, respectively. These two models attempted to determine the minimal total cost and minimize the maximum travel time required by the route flow, and their corresponding goals were to minimize the total construction cost and maximize the area that the designed water supply network could serve. In this study, the cost terms needed to be revised, as presented in Section 4.1. Using knowledge of relevant operations, each model's conditions/constraints and limitations were developed independently. Both linear and non-linear algebra could be used to develop the mathematical model in accordance with the requirements of the model that the researchers intended to build. The community reservoir water supply network design required the addition of the following key components to the model:

(1) The flow of water between two locations may not be possible due to a drainage divide or because the level of the area where the community's reservoir is established is lower than the water demand points.

(2) As the sizes of the reservoirs vary, the model must incorporate the size of the community reservoirs. Considering the appropriate land size or topography at a particular site, reservoir size can render some areas unsuitable for establishment. Therefore, it is imperative that this constraint be introduced into the proposed model.

(3) Gravity flow is utilized to transport water from reservoirs to water demand nodes in the model. Consequently, the maximum distances of the pipes connecting to the reservoir must be enforced so that water can effectively flow from the reservoir to the demand node.

These three restrictions needed to be introduced into the formulation of the mathematical model in addition to those described by Chandra, S. et al. [63] and Demir, I. et al. [64]. The mathematical model could be solved to optimality using exact methods that yielded the optimal solution or heuristic approaches that could obtain a promising solution (albeit one that was not guaranteed to be optimal). While an exact technique can locate the ideal solution, it requires extensive processing time, which grows exponentially as the problem size increases. With a typical personal computer, resolving a capacitated location–sizing–allocation problem with less than one hundred demand points could take up to one month. Therefore, a heuristic approach needed to be developed to solve the proposed model.

The second phase of model building consisted of developing an effective heuristic for solving the mathematical model proposed in the first phase. As noted in the introduction, the suggested model was solved using A-VaNSAS, which was adapted from the variable neighborhood technique adaptive search (VaNSAS). VaNSAS consists of four general processes: (1) the initial set of tracks is generated; (2) the tracks select the improvement approach to enhance the quality of the solution; (3) the heuristics information is updated; and (4) steps (2) and (3) are continued until the termination requirements are met. In the proposed method, VaNSAS was improved in order to increase its search capability. The following details of VaNSAS were modified for A-VanSAS:

(1)   Step 2 of VaNSAS was updated to allow the current best tract to guide the search space.
(2)   Instead of employing three improvement approaches to enhance the solution quality of the tract solutions as in VaNSAS, we utilized five improvement strategies.
(3)   The newly constructed decoding method extracted the solution to the provided mathematical model. Pitakaso, R. et al. [7] proposed a decoding approach for solving the green 2-echelon location routing problem, whereas in this study, we solved the location–sizing–allocation issue. As mentioned in the section on mathematical model formulation, these two types of problems have different model attributes and characteristics; hence, the decoding method from real numbers was required to be the suggested model solution (this is explained in the following section).

*3.2. Model Testing Phase*

In this section, the numerical example of the case study is examined and analyzed. The proposed model and methods for solving the model are disclosed in order to evaluate their efficacy in answering the study objectives. The high-drought-risk area in the city of Khong Chiam, Ubon Ratchathani (see Figure 1), was used as a case study in order to evaluate the effectiveness of the model. Khong Chiam is located in the Mun Basin and is the final Thai city before the waters of the Mun River flow into the Mekong. In this research, we identified the optimal placements for community reservoirs that could lessen the risk of drought in the target area. We then built the water distribution network for lower-risk regions. The goal was to minimize the total cost of construction while maintaining the lowest possible risk of drought in the target area.

The proposed methods were compared with existing heuristics, including differential evolution (DE), the genetic algorithm (GA), and the original VaNSAS. All of the methods were programmed in Python and tested on an Intel(R) Core (TM) i7-7500U CPU with two cores and four logical processors running at 2.70 GHz. All of the proposed methods were tested in a real-world scenario including 218 nodes representing potential water reservoir locations. The computational duration was set to 30 min as is the termination condition of A-VaNSAS, DE, and GA. Khong Chiam City, Ubon Ratchathani Province, Thailand, which is located in the Mun River Basin, was the case study area that was tested using all the methodologies. Data and information were gathered from each node, consisting of $x$ and $y$ coordinates, the height above sea level, and the average crop water requirements (CRW). Approximate construction costs for water reservoirs, irrigation systems, and water receiving systems were computed following the criteria for calculating the median prices of irrigation construction. Three sizes of agricultural water reservoirs were considered: small, medium, and large. The capacity and irrigation distance limitations of each reservoir varied with size. The ranges of the values for the parameters used in the case study are given in Table 1.

**Table 1.** Ranges of the parameters used in the case study.

| Parameters | Range of Value | Unit Name | Parameters | Range of Value | Unit Name |
|---|---|---|---|---|---|
| I | 218 | Locations | $s$ | 0.84 | Baht/m$^2$ [300] |
| K | 3 (SC, LC, AW) | Types | $r_i$ | [0.25, 0.95] | - |
| $f_k$ | [43.47, 12.14, 22.5] | Thousand Baht | $m_k$ | [30, 80, 100] | km |
| J | 218 | Locations | G | 5760.9 | Thousand cubic meters |
| $d_{ij}$ | [0, 15,000] | Meters | $U_k$ | [1260, 3780, 4520] | Thousand cubic meters |
| $C_{ij}$ | [28, 56] | Baht/meters | $w_j$ | [0.04, 169.21] | Thousand cubic meters |
| $n_i$ | [8000–160,000] | m$^2 \times 10^3$ | | | - |

## 4. Results and Discussion

Implementing the research procedures described in Section 3 helped us to answer the research questions. This section presents the model and method proposed to solve the problem of high-drought-risk locations. There are three major components to the results: (1) model development, (2) model validation, and (3) model discussion.

*4.1. Model Building*

The following subsections detail the mathematical model formulation and the development of A-VaNSAS.

4.1.1. Mathematical Model Formulation for Establishment of the Community Reservoir and Water Supply Network Design (CR–WSND)

This section introduces the CR–WSND mathematical model for agricultural applications. To analyze the benefit of establishing CRs in a high-risk region, the objective function was used to reduce the building costs associated with the community reservoirs and network. Figure 3 is a graphical representation of the cost terms in the objective function.

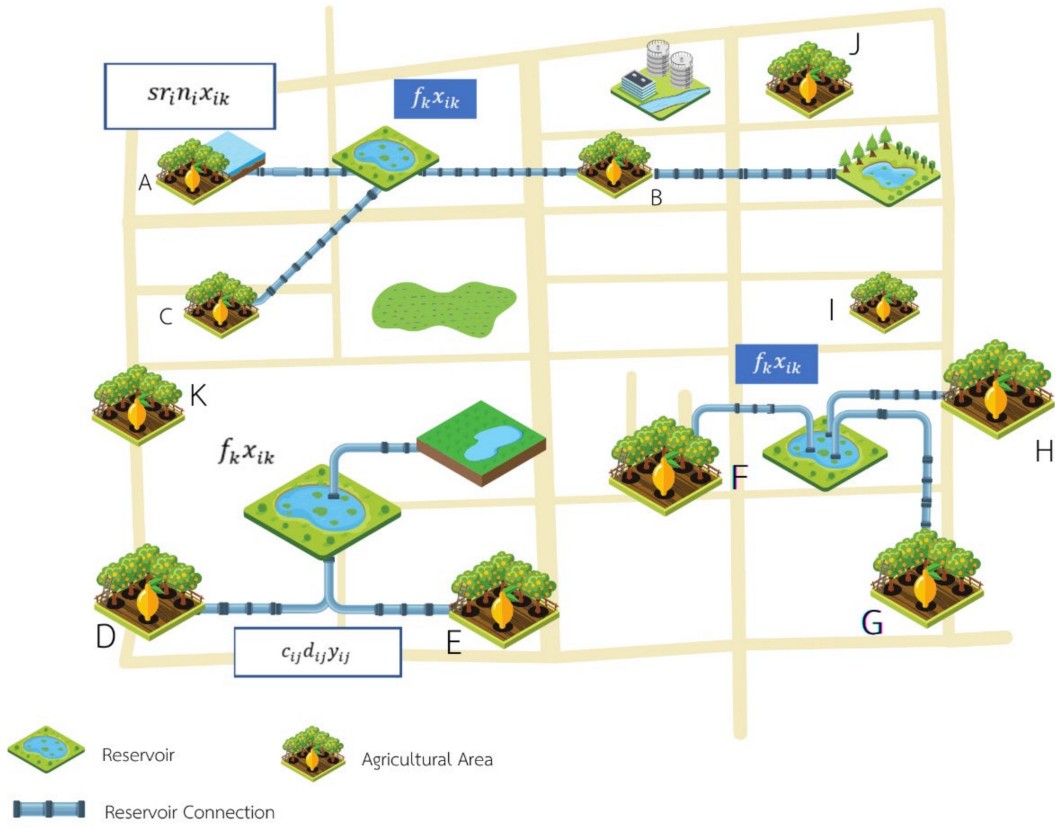

**Figure 3.** Graphical illustration of the proposed mathematical model.

We define the indices as follows: (1) $i$ denotes the candidate areas to locate the CRs, $I = 1, \ldots, I$; (2) $j$ denotes the agricultural areas or water demand nodes, $j = 1, \ldots, J$; and (3) $k$ denotes the types of CRs, $k = 1, \ldots, K$. As Thailand's weather conditions are well known with respect to its heavy rain, the great intensity and variability of the precipitation levels were assumed sufficient to fill all the reservoirs constructed.

Figure 3 is a graphical illustration of the mathematical model, including a set of candidate locations to establish the CRs ($j = 1, \ldots, J$) and a set of candidate types/sizes of CRs ($k = 1, \ldots, K$). We located the different types of CR to generate the associated construction cost ($f_k$). $X_{ik}$ is a binary decision variable, which is equal to 1 if location $i$ is selected to construct a reservoir of type $k$, and 0 if it is not; $f_k$ is the fixed construction cost, which corresponds to the size/type of CR that is selected to be built in area $i$. Equation (1) is used to calculate $f_k$. Denote by $Z_k^1$ the unit cost per square meter of CR type $k$ and by $Z_k^2$ the volume of CR type $k$ (in square meters).

$$f_k = Z_k^1 \, Z_k^2 \,. \tag{1}$$

By multiplying $f_k$ by $X_{ik}$, we obtain the construction cost ($Z_{ik}^3$) for area $i$:

$$Z_{ik}^3 = f_k X_{ik}. \tag{2}$$

$Y_{ij}$ denotes the presence of a water supply network distributing from location $i$ to location $j$, and $C_{ij}$ is the construction cost of the water supply network from nodes $i$ to $j$ (THB/km), including the construction of the water receiver. $Z_{ij}^4$ is the construction cost of the water supply network and $d_{ij}$ is the distance from node $i$ to node $j$. Equation (3) is used to calculate $Z_{ij}^4$ :

$$Z_{ij}^4 = c_{ij} d_{ij} Y_{ij}. \tag{3}$$

The construction cost is defined as the first two cost terms, while the drought risk mitigation incentive (DRMI) is the third term in the objective function. This is a term used in the incentive function to increase the likelihood of establishing a reservoir in a high-drought-risk zone, and the danger varies depending on the candidate area ($r_i$). The chosen location for the CR ($X_{ik}$) should be in an area with a higher risk of drought in order to lessen the likelihood of that area experiencing a drought problem.

Next, $s$ is specified as the subsidiary cost per aridity risk unit (THB/m$^2$); this is the cost that will be activated when the drought occurs. In this case, the government must compensate farmers for agricultural products that have been damaged as a result of the drought. Furthermore, $n_i$ is defined as the agricultural area of location $i$ and $Z_{ik}^5$ is the incentive profit of location $i$ with a type $k$ community reservoir or the so-called drought risk mitigation incentive:

$$Z_{ij}^5 = s r_i n_i X_{ik}. \tag{4}$$

As the goal of solving the CR–WSND is to have the lowest system cost, the objective function is the sum of Equations (2)–(4), stated in Equation (5):

$$Min\ Z = \sum_{k=1}^{K} \sum_{i=1}^{I} Z_{ik}^3 + \sum_{j=1}^{J} \sum_{i=1}^{I} Z_{ij}^4 - \sum_{j=1}^{J} \sum_{i=1}^{I} Z_{ij}^5 . \tag{5}$$

Decision variables $Y_{ij}$ and $X_{ik}$ will be revealed under certain limitations, such as (1) the maximum distance of the water supply network from the selected location to CRs in other locations; (2) the total water supply to other locations in a certain network must be under the maximum capacity of the CR; and (3) the total budget constraints. The parameters used to formulate the constraints are as follows:

**Parameters**

| | |
|---|---|
| $f_k$ | Cost of constructing agricultural water resources with size $k$ (THB/cubic meters) |
| $U_k$ | Volume of available water for CR of size $k$ (m$^3$) |
| $v$ | Cost of constructing irrigation system per distance from agricultural water resource $i$ to demand node $j$ (THB) |
| $w_j$ | Water requirement at node $j$ (m$^3$) |
| $b_{ij}$ | $\begin{cases} 1 & \text{if water from node } i \text{ can have gravity flow to } j \text{ (drainage divide and obstacle)} \\ 0 & \text{otherwise} \end{cases}$ |
| $d_{ij}$ | Distance from agricultural water resource $i$ to demand node $j$ (m) |
| $c_{ij}$ | Construction cost of linking nodes $i$ and $j$ (THB/meters) |
| $m_k$ | Maximum distance of water flow from water resource with size $k$ (m) |
| $r_i$ | Aridity risk in area of node $i$ |
| $G$ | Amount of water supply required by the target area population |
| $O_{ik}$ | $\begin{cases} 1 & \text{if reservoir type } k \text{ can be located at location } i \\ 0 & \text{otherwise} \end{cases}$ |

**Decision variables**

$X_{ik}$ $\begin{cases} 1, & \text{if node } i \text{ is selected to be an agricultural water resource with size } k \\ 0, & \text{otherwise} \end{cases}$

$Y_{ij}$ $\begin{cases} 1, & \text{if water from node } i \text{ is assigned to demand node } j \\ 0, & \text{otherwise} \end{cases}$

**Constraints:**

$$\sum_{j=1}^{J} Y_{ij} w_j \leq \sum_{k \in K} U_k X_{ik} \qquad i \in I \qquad (6)$$

$$\sum_{k \in K} X_{ik} \leq 1 \qquad i \in I \qquad (7)$$

$$d_{ij} Y_{ij} \leq \sum_{k \in K} m_k X_{ik} \qquad i \in I, j \in J \qquad (8)$$

$$\sum_{j \in J} Y_{ij} \geq \sum_{k=1}^{K} X_{ik} \qquad i \in I \qquad (9)$$

$$Y_{ij} h_j \leq h_i \qquad i \in I, j \in J \qquad (10)$$

$$Y_{ij} \leq b_{ij} \qquad i \in I, j \in J \qquad (11)$$

$$\sum_{i \in I} \sum_{k \in K} U_k X_{ik} \geq G \qquad (12)$$

$$\sum_{i \in I} Y_{ij} \geq 1 \qquad j \in J \qquad (13)$$

$$X_{ik} \leq O_{ik} \qquad i \in I, k \in K \qquad (14)$$

Constraint (6) is used to ensure that the water supply from node $i$ is a sufficient volume of water to supply its entire network. Constraint (7) is used to limit each water resource $i$ to only have one size $k$. Constraint (8) is used to ensure that the distance of the water supply from node $i$ to node $j$ does not exceed the maximum distance of water flow from a water resource of size $k$. Constraint (9) is used to ensure that the water flow from $i$ to $j$ occurs only when $i$ establishes at least one reservoir. Constraint (10) is used to ensure that the water flows from a higher altitude to a lower altitude; for example, if location $a$ has a height 148 m above sea level and location $b$ has a height 190 m above sea level, the restriction in this constraint is that $Y_{ab}$ cannot be 1, as $b$ is higher than $a$ and, thus, gravity water flow from $a$ to $b$ is not possible. Constraint (11) is used to limit the flow if a drainage divide or another water flow obstacle blocks the waterway from node $i$ to node $j$.

The MIP decides to site the CR at position 1 and create a network from (1) to (2); however, this is not possible due to the drainage divide between (1) and (2). Connecting (1) to (2) is difficult. Therefore, the value of $b_{12}$ is set to be zero. $b_{ij}$ are pre-defined parameters derived from the real landscape of the target area. If a pip cannot be installed between locations $i$ and $j$, the value of $b_{ij}$ for that connection will be set to zero. Constraint (12) guarantees that the total amount of water gathered from all the CRs exceeds the total amount of water required by all the demand nodes. Constraint (13) ensures that each water demand node is served by at least one CR, whereas Constraint (14) ensures that only those locations that are suitable for locating reservoir type k are allowed to do so.

### 4.1.2. Model Testing

In this section, we evaluate the efficacy of the proposed model and method. An improved version of the variable neighborhood strategy adaptive search (A-VaNSAS) is introduced in this study. A-VaNSAS has four processes: (1) creating initial tracks (set of WPs); (2) running the track touring process; (3) updating the heuristics information; and (4) repeating steps (2) and (3) until the system terminates. Before explaining the proposed method (A-VaNSAS), the case study information utilized to test the model and method is described in the following section.

### Case Study Data for Community Reservoir and Water Supply Network Design

In this section, the process followed to design the community reservoirs is discussed. We designed three sizes of clay pond as the reservoirs, and all the community reservoirs used a gravity flow water distribution system. This is a system in which the water is provided by gravity flow such that the distribution reservoir must be located at a higher

elevation than the target community. When the source is a river or an impounded reservoir at a sufficient height above the target settlement, this method is considerably more appropriate. The advantages of this system are as follows: (1) the system requires no energy to operate, as the water is transported by gravity; (2) it is not necessary to have a pump; and (3) it is cost-effective in the long term.

The small, medium, and large clay ponds were modified from the model presented by the Land Development Department, Ministry of Agriculture and Cooperatives, Thailand. A diagram of the water flow system and the pressure conditions in the water supply system is provided in Figure 4. The small, medium, and large clay ponds had the capacities for water retentions of 1260, 3780, and 4520 cubic meters, respectively. Details regarding the sizes of the small, medium, and large clay ponds are given in Table 2.

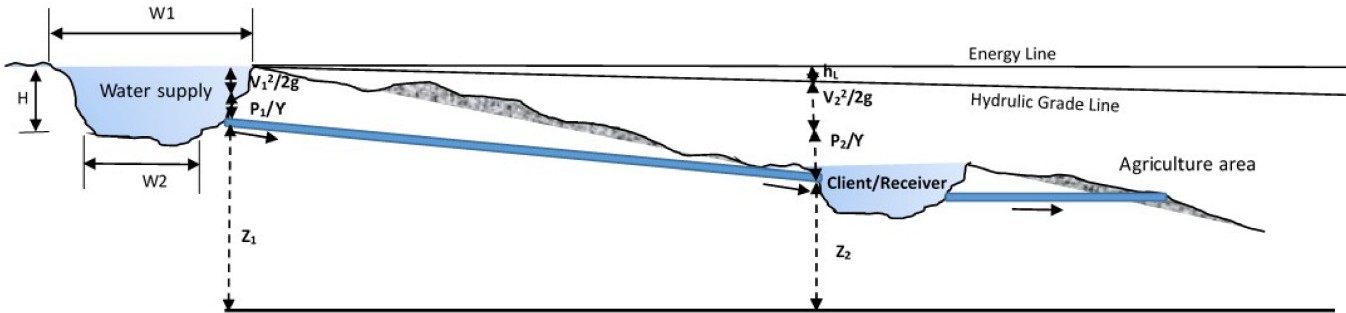

**Figure 4.** Diagram of the gravity flow system of the designed water supply.

**Table 2.** Details of small and large community reservoirs.

| Parameter | Small | Medium | Large |
|---|---|---|---|
| Capacity (cubic meters) | 1260 | 3780 | 4520 |
| W1 (meters) | 19.0 | 19.0 | 19.0 |
| W2 (meters) | 15.0 | 15.0 | 15.0 |
| L1 (meters) | 39.0 | 39.0 | 39.0 |
| L2 (meters) | 35.0 | 35.0 | 35.0 |
| H (meters) | 2.0 | 6.0 | 7.2 |
| Maximum distance of water Network pipe (meters) | 50,000 | 100,000 | 150,000 |
| Construction cost (thousand baht) | 43,470 | 121,440 | 255,000 |

In Figure 4, Bernoulli's principle is a seemingly counterintuitive statement with which to describe the pressure conditions in the water supply system; $Z_i$ is the elevation head, $\frac{P_i}{\gamma}$ is the pressure head, $\frac{v_i^2}{2g}$ is the velocity head, and $h_L$ is the head loss.

In the specified water supply system, gravity watering was considered since it does not require any energy to convey water through a pipeline and generate pressure at the trough. The energy due to gravity at a location is equal to the difference in elevation between sites, such as between the water supply and the trough. The term "head" is used to represent this difference in altitude in meters. In the case of water, this energy is comparable to either 0.3048 m of height drop, which corresponds to 0.433 psi of pressure head, or 0.704 m of height drop, which corresponds to 1 psi of pressure head [65]. The water pressure in the trough is determined by the pipe size, the pipe material, and the water supply of the watering system. Different pipe sizes and pipe materials have varying flows for a given change in elevation due to their varying friction losses. Through the translation of energy into heat, the friction losses are exclusively lost to the water system.

As indicated previously, water flow pressure is also dependent on the quantity of water supply reservoirs, and seasonal variations may occur. For example, the constructed

reservoirs' water sources were groundwater and precipitation. Water was supplied to stand posts and agricultural area connections from the reservoir. In order to maintain the steady flow, all the troughs needed to be of sturdy construction. To design a pipeline, companies must conduct a site inspection, choose a route, and, if necessary, examine the location of the valves and the break pressure area. The required pipe size should be determined by an engineer based on the elevation difference and anticipated flow rates (in our case, the details of the designed water supply system are shown in Figure 3). Valves and pipe fittings were used for controlling the water flow. Taps and valves were necessary for creating connections between the agricultural fields and stand posts.

The clay ponds had varying construction costs due to their different sizes; the cost of building a small, medium, and large clay pond was THB 43,470, THB 121,400, and THB 255,000, respectively.

The demand for water from agricultural areas is determined by the amount needed to produce each type of agricultural product in that region (in terms of daily water used). The water that flows into built reservoirs from both rain and groundwater will flow out proportionally to the demand that each reservoir serves. Thus, the size of a reservoir to be be constructed in a given region must be determined according to the demand of all the regions that it must serve.

These three types of community reservoirs were chosen for the placement and distribution of water to the desired areas. Due to their limited capacity, it is only possible to supply water to a restricted number of sites. Thus, the considered problem is composed of three sub-problems, collectively known as the location—-sizing–allocation problem. While deciding on the locations of community reservoirs, it was necessary to identify the sizes of the reservoirs to be used. Finally, a water supply network also needed to be built in order to meet the needs of neighboring towns, which was referred to as the location–sizing–allocation dilemma. The data provided in this section are used in the subsequent section in order to evaluate the proposed model and the efficacy of the suggested approach, that is, A-VaNSAS.

Adjusted Variable Neighborhood Strategy Adaptive Search Solving Community Reservoir and Water Supply Network Design

The steps for using A-VaNSAS to solve the CR–WSND problem are outlined below. The track for the depiction of the CR—-WSND was designed as a $1 \times D$ track in the production of a set of starting tracks, where NT was the number of tracks. For example, when D equaled 10, the track was composed of 10 real numbers, which were 0.67, 0.43, 0.03, 0.10, 0.69, 0.72, 0.54, 0.56, 0.58, and 0.68. This number was decoded to obtain the result of the proposed problem, which is explained below. An example of five randomly selected tracks (NP = 5) is given in Appendix A.

The decoding procedure was divided into nine steps: (1) sort the values in the WP's position in ascending order; (2) choose the value node with the lowest value as the water reservoir; (3) determine the reservoir sizing criteria by assigning the same probability to each reservoir size; (4) as a result of (2), generate a random number for the node; (5) based on this new random value, determine the reservoir size using the criteria stated in step (3); (6) sum the water demand from all the nodes as denoted by T; (7) repeat steps 2–5 until the total capacity of all the established reservoirs exceeds 1.2 T; (8) assign the remaining demand nodes to the selected water reservoirs; and (9) repeat steps (8) and (9) until all the reservoirs have been assigned.

The demand nodes should be assigned to the first established reservoir until it runs out of capacity, at which point the remaining demand nodes are sent to the next established reservoirs. The following constraints must be considered when assigning the network: (1) the distance limitation for irrigation; (2) the crop water requirements of the assigned demand nodes, which must not exceed the capacity of water reservoirs in supply nodes; and (3) the height above sea level of the supply nodes, which must be greater than that of the demand nodes. Example supply and demand node information is provided in Appendices B and C. The results of the assignment method are shown in Table 3.

**Table 3.** Results of the assignment method.

| Water Reservoir | Size | Supply to Node | CR Construction Cost (baht) | Water Network Construction Cost (baht) | DRMI (baht) | Total Cost (baht) |
|---|---|---|---|---|---|---|
| 3 | AW | 3,8,9,1,10 | 255,000 | 95,700 | 10,900 | 339,800 |
| 4 | SC | 4 | 43,470 | 0 | 7875 | 35,595 |
| 2 | SC | 2 | 43,470 | 0 | 8559 | 34,911 |
| 7 | LC | 7,5,6 | 121,440 | 52,890 | 10,060 | 164,270 |
| | Grand total cost | | 463,380 | 148,590 | 37,394 | 574,576 |

From Table 3, nodes 2, 3, 4, and 7 were selected to locate the water reservoirs. Node 3 had the lowest value in position and a probability of 0.78; thus, AW was selected as the size of node 3. The probabilities for nodes 2, 4, and 7 were 0.04, 0.21, and 0.54, respectively, such that SC, SC, and LC were the selected types of each node, respectively. First, node 3 needed to satisfy its own water supply (928.1 cubic meters), following which nodes 8, 9, 1, and 10 were assigned to node 3, with a total water demand of 4132.7 cubic meters. The construction cost of the AW reservoir was THB 255,000, and the water network construction cost was THB 52/meter. In this scenario, node 3 distributed its water over a total distance of 1840.38 m to nodes 8, 9, 1, and 10, resulting in a total construction cost of THB 95,700. The government compensates farmers who are afflicted by drought disasters in their area through a drought risk mitigation incentive. Drought subsidy rates have recently averaged THB $0.87/m^2$. The total drought risk mitigation incentive is determined by a node's drought risk, which is estimated based on the likelihood of drought in the area. As node 3 had an area of 12,528.7 $m^2$ and a drought risk of 0.85, the drought risk mitigation incentive for node 3 was equal to THB 10,900. Reservoirs 4, 2, and 7 were analyzed using the same approach.

The tracks then iteratively toured the black box throughout the track touring process. A black box—also known as an improvement box—contains strategies for improving solutions that are not limited to local searches. Metaheuristics, heuristics, basic local searching, and other techniques may be included in a black box. Five black boxes were created for this study. To choose the preferable black box, a roulette wheel was applied to select them for the track. Equation (15) controlled the chance of picking the black box for the original VaNSAS, while Equation (16) was used for the adjusted VaNSAS (A-VaNSAS).

$$P_{bt} = \frac{FN_{bt-1} + (1-F)A_{bt-1} + KI_{bt-1}}{\sum_{b=1}^{B} FN_{bt-1} + (1-F)A_{bt-1} + KI_{bt-1}}, \tag{15}$$

$$P_{bt} = \frac{FN_{bt-1} + (1-F)A_{bt-1} + KI_{bt-1} + \rho\left|A_{bt-1} - A_{t-1}^{best}\right|}{\sum_{b=1}^{B} FN_{bt-1} + (1-F)A_{bt-1} + KI_{bt-1} + \rho\left|A_{bt-1} - A_{t-1}^{best}\right|}, \tag{16}$$

where $P_{bt}$ is the probability of selecting the black box in iteration $t$; $N_{bt-1}$ is the number of tracks that selected a black box in the previous iteration; $A_{bt-1}$ is the average objective value of all the tracks that selected a black box in the previous iteration; $A_{t-1}^{best}$ is the average objective value of the best black box in the iteration $t$; $I_{bt-1}$ is the reward value, which increases by 1 if a black box finds the optimal solution in the last iteration, but is set to 0 otherwise; $IB$ is the total number of black boxes; $F$ is the scaling factor ($F = 0.5$); and $K$ is the parameter factor ($K = 0.3$). The black boxes used in this study were as follows: (1) THE swap method; (2) 2-opt method; (3) K transition method; (4) K cyclic move method; and (5) G-best transition method.

The swap method is the classic method for improving the quality of the solution. First, two tracks are randomly selected. Then, one node from each track is chosen and swapped. The 2-opt method is similar to the swap method, but the swap operations occur only in the same track.

The K transition method (KTM) allows the track to move away from local optima by transforming some values in some nodes to random new values. KTM consists of four steps: (1) the random selection of candidate tracks; (2) the selection of random value K, representing the number of transformed values; (3) the selection of K nodes in the target track randomly; and (4) the transmission of values to the new candidate value or new selected track.

The K cyclic move method (KCM) was also designed to improve the quality of solutions. This method includes four processes: (1) Randomly select the move factor K; (2) randomly select the K tasks; (3) cyclically move according to the sequences selected in step (2); and (4) carry out steps 1–3 until a pre-defined number of iterations is reached.

The G best transition method (GBT) conducts a search by using the best tracks from a previous search. The steps of GBT are as follows: (1) randomly select value G; (2) select the K points randomly for the transmission of the node values; (3) swap the value(s) from the selected best track to the target track; and (4) carry out steps (1)–(3) until a stopping condition is met. Examples of KTM, KCM, and GBT are shown in Appendices D–F.

The result from the decoded solution for the new selected track was applied in the next iteration of A-VaNSAS. The heuristic information must be updated when all the tracks finish their process in the black box. The probability of the black box was updated and the track touring process was repeated using Equations (17) and (18):

$$A_{bt} = \frac{N_{bt}}{T_{bt}},$$ (17)

$$I_{bt} = I_{bt-1} + G,$$ (18)

where $N_{bt}$ is the total number of tracks which select black box $b$ from iteration 1 to $t$; $T_{bt}$ is the total cost generated from all the tracks that select black box $b$ from iteration 1 to $t$; and $G$ is 1 if black box $b$ contains the global best solution in iteration $t$ and is 0 otherwise. The pseudocode of A-VaNSAS is shown in Appendix G.

In addition, the effectiveness of A-VaNSAS was compared to that of the genetic algorithm (GA) and differential evolution (DE) algorithm. A genetic algorithm is a four-step metaheuristic inspired by nature [66], while differential evolution is a metaheuristic method for solving problems that involves five steps: (1) generate a new track randomly; (2) execute a mutation process; (3) perform a re-combination process; (4) perform a selection process; and (5) repeat steps (2)–(4) until the termination condition is met [67].

The results from testing GA, DE, VaNSAS, and A-VaNSAS via simulation are given in Table 4.

**Table 4.** Computational results of the case study.

| KPI | Algorithm | | | |
|---|---|---|---|---|
| | **GA** | **DE** | **VaNSAS** | **A-VaNSAS** |
| NC (nodes) | 218 | 218 | 218 | 218 |
| TDWS (cubic meters) | 419,248 | 419,248 | 419,248 | 419,248 |
| TNCR (nodes) | 136 (s = 43 m = 70 l = 23) | 131 (s = 35 m = 81 l = 22) | 142 (s = 51 m = 62 l = 29) | 128 (s = 28 m = 90 l = 10) |
| TVW (cubic meters) | 422,740 | 449,720 | 429,700 | 420,680 |
| TVW/TDWS | 1.01 | 1.07 | 1.02 | 1.00 |
| TCC (Baht) | 19,941,410 | 20,584,090 | 20,576,450 | 18,764,760 |
| TDRC (Baht) | 5,266,007 | 5,501,719 | 5,626,589 | 5,769,953 |

KPI: key performance indicators; NC: number of candidate nodes; TDWS: total demand of water supply; TNCR: total number of CRs constructed; TVW: total volume of water able to be collected by all CRs; TCC: total construction cost; TDRC: total drought risk mitigation incentive.

Table 4 shows that A-VaNSAS surpassed the original VaNSAS in terms of its computational results. It cut construction costs by 12.62% compared with those of the original VaNSAS and by 7.78% and 14.19% compared with those of GA and DE, respectively. A-VaNSAS provided a 5.56%, 10.73%, and 2.10% higher drought risk mitigation incentive (DRMI) than GA, DE, and VaNSAS did, respectively. Thus, A-VaNSAS can reduce the risk of drought more effectively than other techniques. Based on the computational results, A-VaNSAS required THB 18,764,760 to install all of the community reservoirs and their network to protect the community from the drought situation and meet the demand for water from all the demand nodes. Every year, the government funds a portion of the cost that are required to build reservoirs. However, as indicated, they are only able to create reservoirs in a few areas. Figure 5 provides a graphic representation of the drought risk areas, indicating their drought risk reduction once the reservoirs are built. In the target region, 100% of the areas in the highest risk category were protected.

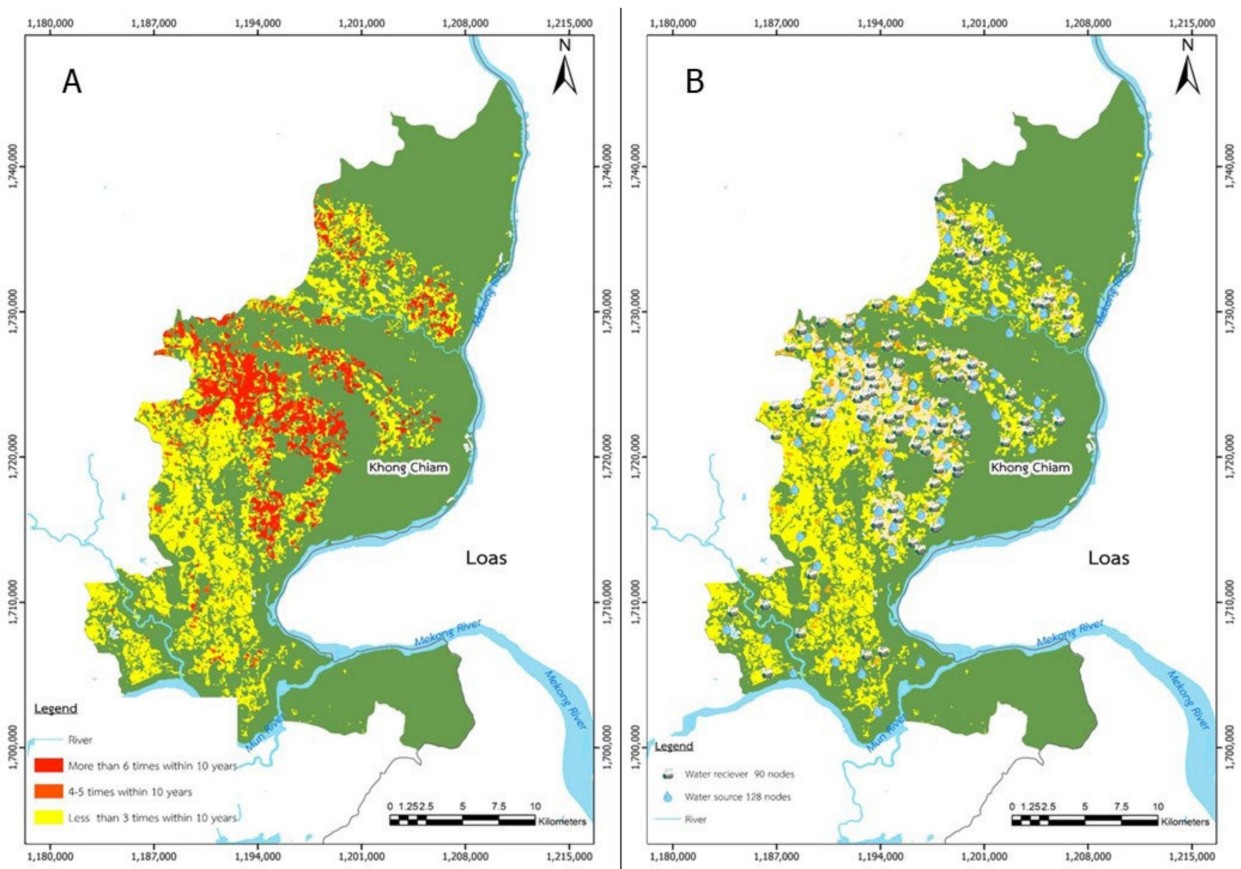

**Figure 5.** Comparison of drought risk in Khong Chiam (**A**) before and (**B**) after CR placement.

The limited budget was included in the model for this experiment, which had an obvious impact on the solution. The MIP provided in Section 4 required the following calculated adjustments: $P$ was the budget available for the current year, and $Z_{ik}^3$ and $Z_{ij}^4$ denoted the construction costs (Equations (2) and (3)). Constraint (19) was used to control the total construction cost such that it did not exceed $P$. Thus, Constraint (19) was added into the model detailed in Section 4.

$$\left( \sum_{k=1}^{K} \sum_{i=1}^{I} Z_{ik}^3 + \sum_{j=1}^{J} \sum_{i=1}^{I} Z_{ij}^4 \right) \leq P. \tag{19}$$

Since Constraint (19) was added into the model, Constraints (12) and (13) needed to be adjusted. Constraint (12) ensured that the water supply requirements of all the nodes

were met. Since the budget was limited, it was possible that some nodes would be left out of the water supply network. With the limited budget, Constraint (13) could also be violated. Although Constraints (12) and (13) were removed, the objective function Equation (5) was changed to ensure that the model searched for the lowest cost while simultaneously attempting to reduce the number of places that were not served by the water network, as follows:

$$Min\ Z = \sum_{j=1}^{J} \sum_{i=1}^{I} Z_{ij}^6, \tag{20}$$

where $Z_{ij}^6$ is defined as:

$$Z_{ij}^6 = sn_i r_i (1 - Y_{ij}) \tag{21}$$

$Z_{ij}^6$ was the value of risk in the area that was not integrated into the water supply network. The experiment was carried out on the case study area by cutting a limited figure from the initial budget, that is, cutting the initial budget by 10%, 20%, 30%, 40%, or 50%. The percentage of overall risk in the area not covered by water supply was divided by the total risk value ($sn_i r_i$) of all the locations (RS; Equation (22)). The Total number of sites in the water supply network (LW), as well as the total volume of water that could be collected in each scenario, was divided by the total demand for water from all the nodes (WS; Equation (23)).

$$RS = \frac{\sum_{j=1}^{J} \sum_{i=1}^{I} Z_{ij}^6}{\sum_{i=1}^{I} sn_i r_i} \times 100\%, \tag{22}$$

$$WS = \frac{\sum_{i=1}^{i} \sum_{k=1}^{K} m_k X_{ik}}{\sum_{j=1}^{J} w_j} \times 100\%. \tag{23}$$

When the RS is high, the overall risk of drought is high. As a result, the algorithms needed to optimize for a lower RS. The percentage of the total area serviced by the water delivery network was denoted by the WS; as such, an effective method would produce high WS.

$$\%|diff| = \frac{|V^{25\%} - V^{0\%}|}{V^{0\%}} \times 100\%. \tag{24}$$

The %|diff| of LW was calculated using Equation (24), where $V^{25\%}$ was the value of the LW at a 25% increase in construction costs and $V^{0\%}$ was the base value before the increase. The %|diff| of RS and WS was calculated using Equation (25):

$$\%|diff| = V^{25\%} - V^{0\%}. \tag{25}$$

Table 5 demonstrates that when A-VaNSAS was given the same constrained budget, it delivered the best solution in terms of all the KPIs when compared with the other methods. In the water supply network, A-VaNSAS integrated an average of 166.8 locations, while GA, DE, and VaNSAS could only integrate 155.5, 151.5, and 151.7 locations, respectively. In other words, A-VaNSAS achieved a higher number of places served by the water supply network (6.79−9.09% higher) than the other approaches. In terms of additional KPIs, A-VaNSAS also outperformed the other techniques; namely, it could increase the volume of saved water for use in the system by at least 4.60% and minimize the danger of drought by at least 21.31%.

**Table 5.** Total risk and LW of the proposed method with different budget limitations.

| % Budget Reduced | Available Budget (baht) | GA | | | DE | | | VaNSAS | | | A-VaNSAS | | |
|---|---|---|---|---|---|---|---|---|---|---|---|---|---|
| | | RS | WS | LW | RS | WS | LW | RS | WS | LW | RS | WS | LW |
| 0% | 18,764,760 | 6.8 | 92.3 | 191 | 7.4 | 92.3 | 190 | 5.6 | 94.8 | 204 | 0.0 | 100.0 | 218 |
| 10% | 16,888,284 | 10.7 | 89.0 | 186 | 11.2 | 88.1 | 184 | 9.5 | 90.1 | 193 | 7.1 | 92.3 | 199 |
| 20% | 15,011,808 | 23.5 | 76.4 | 163 | 23.8 | 76.9 | 168 | 18.6 | 81.5 | 166 | 16.6 | 83.8 | 177 |
| 30% | 13,135,332 | 31.8 | 68.8 | 147 | 32.6 | 67.6 | 144 | 30.9 | 69.3 | 142 | 25.3 | 73.5 | 154 |
| 40% | 11,258,856 | 33.6 | 63.6 | 120 | 33.1 | 64.8 | 123 | 33.4 | 65.9 | 126 | 29.9 | 70.1 | 137 |
| 50% | 9,382,380 | 41.8 | 59.4 | 103 | 42.9 | 58.8 | 100 | 42.0 | 58.3 | 102 | 36.5 | 62.4 | 116 |
| average | 14,073,570 | 24.7 | 74.9 | 151.7 | 25.2 | 74.8 | 151.5 | 23.3 | 76.7 | 155.5 | 19.2 | 80.4 | 166.8 |
| % |diff| | | 35.0 | 32.9 | 46.1 | 35.5 | 33.5 | 47.4 | 36.4 | 36.5 | 50.0 | 36.5 | 37.6 | 46.8 |

Units of RS, WS, and LW are percent (%), percent (%), and number of locations, respectively.

Figure 6 shows the percent differences in the KPIs for all the proposed methods with budget reductions ranging from 0% to 50%.

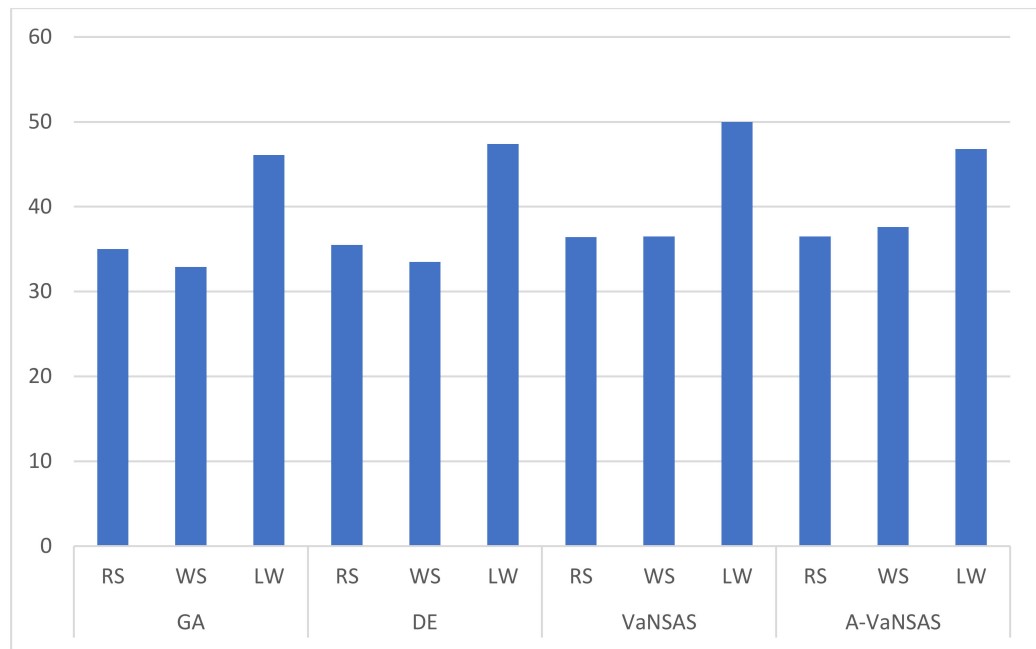

**Figure 6.** Percent differences in KPIs with budget reductions from 0% to 50%.

As the percentage change of LW was the largest of all the proposed solutions, Figure 6 demonstrates that the number of locations that are integrated into the water supply network (LW) was the most sensitive KPI to budget changes. For the results shown in Figure 6, there was not much of a difference between RS and WS. The number of locations established had a larger gap, thus increasing the risk. This can be understood to mean that the proposed algorithms would choose the locations with the largest sizes to lower the overall risk. As the major objective (Equation (20)) is focused on reducing the total risk of the water supply system, the proposed method would not only choose a greater land size but also the locations with the highest risk.

Figure 7 depicts the impact of a 50% reduction to the budget. It shows the original version (with the full budget) and the high drought risk area increasing due to budget constraints. As the price of oil has grown considerably in recent years, so too has the cost of building; therefore, in this experiment, all the KPIs were evaluated under increased construction cost conditions. The percentage increases in building costs considered in this experiment were 5%, 10%, 15%, 20%, and 25%. The same KPIs from the previous trial were recorded during the simulation. Table 6 shows the outcome of the experiment.

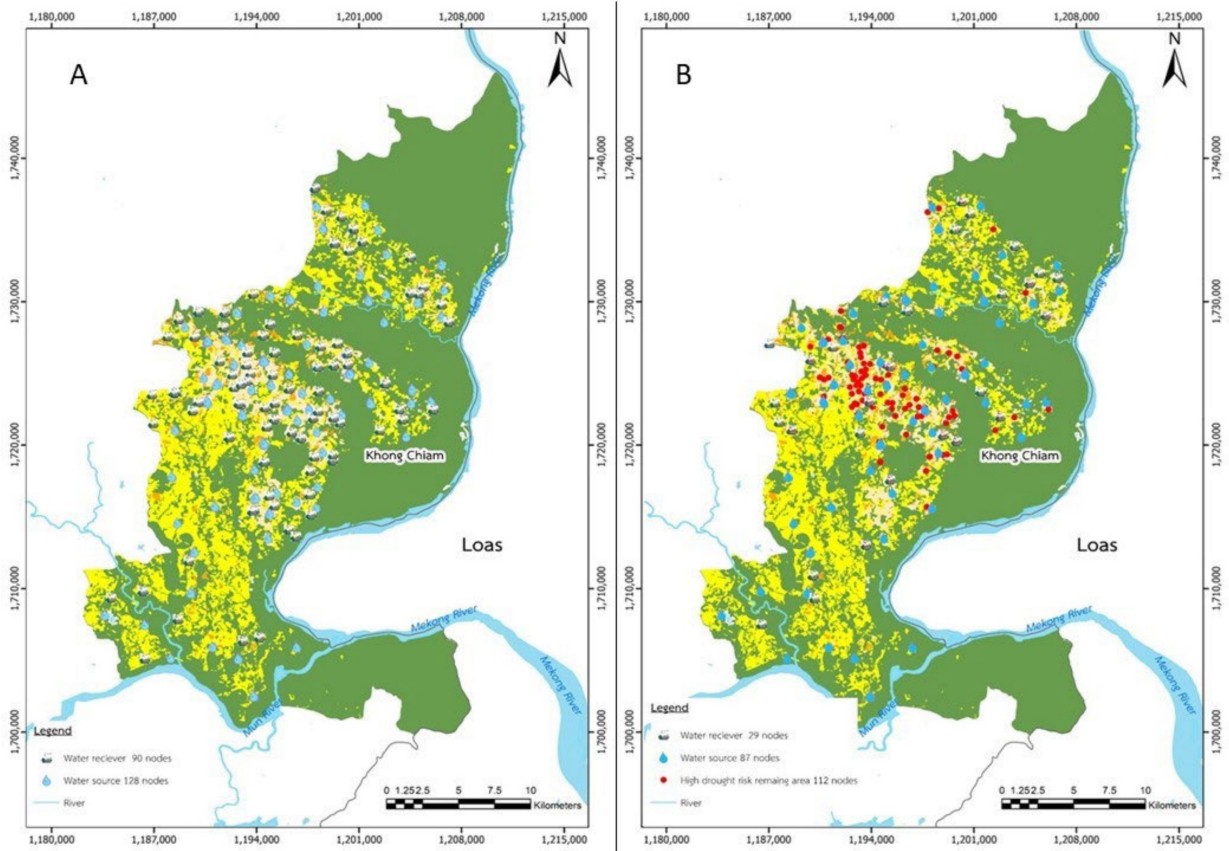

**Figure 7.** Comparison of drought risk in Khong Chiam (**A**) when budget reduction of 0% and (**B**) 50%.

**Table 6.** Total risk and LW of the proposed methods using different construction costs.

| % Construction Cost Increased | GA | | | DE | | | VaNSAS | | | A-VaNSAS | | |
|---|---|---|---|---|---|---|---|---|---|---|---|---|
| | RS | WS | LW | RS | WS | LW | RS | WS | LW | RS | WS | LW |
| 0% | 41.8 | 59.4 | 103 | 42.9 | 58.8 | 100 | 42.0 | 58.3 | 102 | 36.5 | 62.4 | 116 |
| 5% | 44.3 | 56.6 | 94 | 45.4 | 56.1 | 93 | 44.2 | 57.8 | 96 | 38.1 | 60.8 | 109 |
| 10% | 47.5 | 53.8 | 87 | 46.9 | 54.8 | 90 | 46.8 | 55.1 | 92 | 39.8 | 59.3 | 105 |
| 15% | 48.9 | 52.6 | 82 | 47.8 | 54.2 | 88 | 47.3 | 53.7 | 87 | 41.4 | 57.1 | 94 |
| 20% | 51.8 | 50.4 | 79 | 50.1 | 51.3 | 81 | 49.9 | 51.5 | 82 | 43.7 | 56.3 | 89 |
| 25% | 52.5 | 48.2 | 74 | 52.3 | 48.5 | 74 | 51.5 | 49.2 | 76 | 45.6 | 54.9 | 82 |
| Average | 47.8 | 53.5 | 86.5 | 47.6 | 54.0 | 87.7 | 47.0 | 54.3 | 89.2 | 40.9 | 58.5 | 99.2 |
| % \|diff\| | 10.7 | 11.2 | 28.2 | 9.4 | 10.3 | 26.0 | 9.5 | 9.1 | 25.5 | 9.1 | 7.5 | 29.3 |

The average values of RS, WS, and LW for A-VaNSAS surpassed other approaches in terms of finding a better solution, as shown by the computational results in Table 6; GA had lower RS, WS, and LW values than A-VaNSAS by 17.6%, 49.2%, and 3.9%, respectively; DE had lower RS, WS, and LW values than A-VaNSAS by 3.3%, 37.3%, and 11.29%, respectively; and VaNSAS has lower RS, WS, and LW values than A-VaNSAS by 4.4%, 21.3%, and 13.0%, respectively. Figure 8 shows the average values of all the A-VaNSAS KPIs when the construction cost increase was adjusted from 0% to 25%.

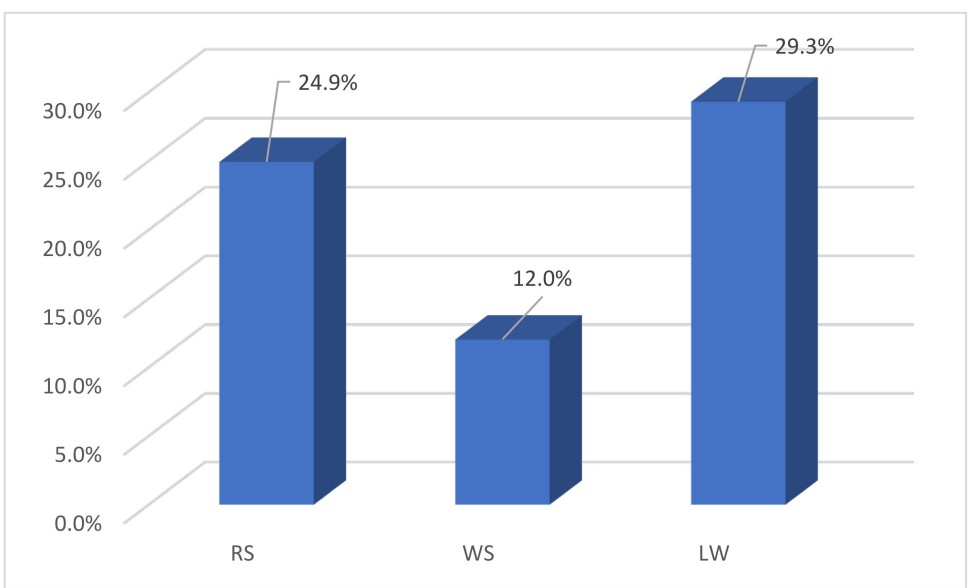

**Figure 8.** Average %|diff| of A-VaNSAS KPIs.

Figure 9 shows the results for the KPI most sensitive to changing construction costs: the number of areas not linked to the water supply network. The percentage of water demand satisfied by the demand node and the drought risk were both affected by changes in construction costs. Table 7 lists the results obtained under a 50% reduction in budget and a 20% rise in construction costs compared with the existing scenario.

Figure 9 compares the risk area of the usual construction cost scenario to a scenario where the construction cost is increased by 50%. Investment into reservoirs was reduced due to the increasing construction costs; thus, the number of high-risk areas covered by reservoirs also decreased.

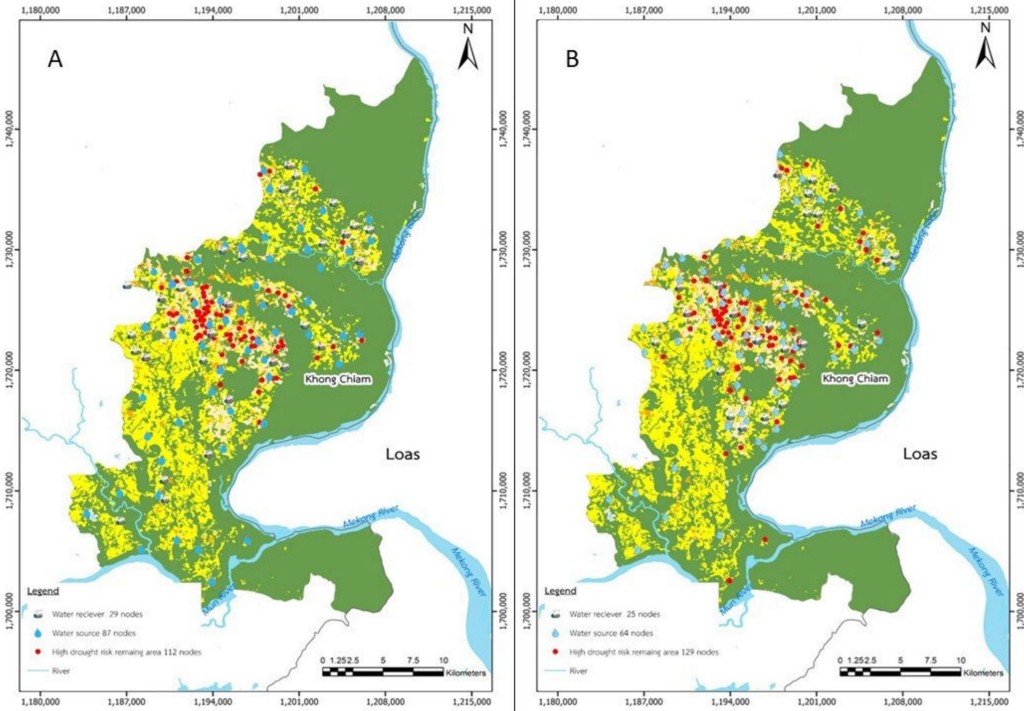

**Figure 9.** Comparison of risk area in Khong Chiam with construction costs increased by (**A**) 0% and (**B**) 50%.

**Table 7.** Comparison of proposed methods under limited resources with respect to all of the KPIs.

| KPIs and Details | GA | DE | VaNSAS | A-VaNSAS |
|---|---|---|---|---|
| Number of candidate nodes: NC (nodes) | 218 | 218 | 218 | 218 |
| Total area available: TA (m$^2$) | 1,217,600 | 1,217,600 | 1,217,600 | 1,217,600 |
| Budget availability: BA (Baht) | 9,382,380 | 9,382,380 | 9,382,380 | 9,382,380 |
| Total demand of water supply: TDWS (cubic meter) | 419,248 | 419,248 | 419,248 | 419,248 |
| Number of locations which are in the water supply network (WSN): NLW (nodes) | 79 | 81 | 82 | 89 |
| Number of locations that construct the community's reservoir: NLC | 60 (s = 12 m = 32 l = 16) | 58 (s = 11 m = 31 l = 16) | 60 (s = 13 m = 31 l = 16) | 64 (s = 13 m = 40 l = 11) |
| Full capacity of the constructed reservoirs (thousand cubic meter) (FCC) | 208,400 | 203,360 | 205,880 | 217,300 |
| (FCC/TDWS) × 100 (%) | 49.7 | 48.5 | 49.1 | 51.8 |
| Total area covered by WSN (m$^2$): TRC | 604,156 | 591,274 | 598,755 | 632,181 |
| Total construction cost (baht): CC | 9,304,720 | 9,311,810 | 9,355,750 | 9,302,710 |
| CC/BA | 99.2 | 99.2 | 99.7 | 99.2 |
| Total drought risk mitigation incentive (baht): SC | 2,367,792.00 | 2,288,865.60 | 2,367,792.00 | 2,525,644.80 |

In the final experiment, GA, DE, VaNSAS, and A-VaNSAS were used to build the required network with just half of the resources available. The results for the above-mentioned KPIs are provided in Table 7.

Table 7 shows that if GA, DE, VaNSAS, and A-VaNSAS were given the same number of resources to manage, A-VaNSAS would be able to provide the best solution with respect to all the KPIs. A-VaNSAS had 89 locations in the WSN compared with 79, 81, and 82 for GA, DE, and VaNSAS, respectively. The A-VaNSAS community reservoirs had a total capacity of 217,300 cubic meters, while the other approaches reserved 4.1–6.4% less water than A-VaNSAS. Figure 10 depicts the investment per square meter required to lower the risk of drought in the case study for all the options. A-VaNSAS used 4.66%, 7.02%, and 8.18% less costs per unit than GA, DE, and VaNSAS, respectively. This suggests that A-VaNSAS was more successful than the other algorithms in reducing the area at risk of drought.

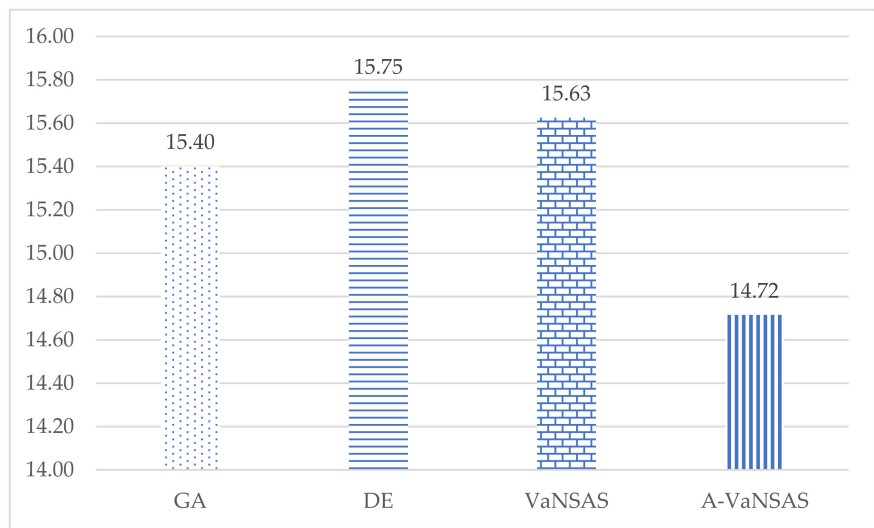

**Figure 10.** Construction cost per square meter to reduce drought risk using various methods (Baht).

Figure 11 depicts the proportion of area where drought risk was mitigated using various methods. A-VaNSAS can reduce the drought risk by 51.92%, whereas VaNSAS, GA, and DE can reduce the drought risk by 49.18%, 48.56%, and 49.62%, respectively. This

suggests that A-VaNSAS was superior to all the other strategies with respect to decreasing the drought-prone area.

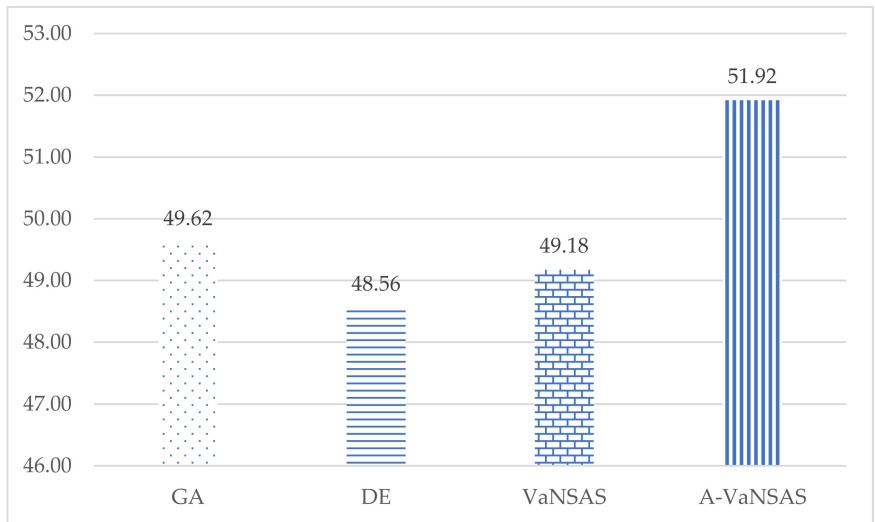

**Figure 11.** Percentage of drought risk area using various solution approaches.

*4.2. Discussion*

Figure 7 and Table 7 illustrate that the water supply model created for this study can lower the probability of drought and its detrimental impact on agricultural production. This result implies that reservoirs or small reservoirs connected to a community pipe water supply network can effectively mitigate drought risk in high-risk areas, which is in agreement with the outcomes reported in [37–40]. As a consequence of these studies, it was concluded that Ethiopia, Australia, and Brazil, among other nations, could benefit from the use of small reservoirs to combat drought, which would include not only the construction of community reservoirs, but also the development of pipe water supply networks that connect the reservoirs to other locations. Our results indicate that the proposed model can effectively mitigate drought in these regions if the community reservoirs and their networks are positioned in the optimal locations. The findings of Collischonn, B. et al. [42] confirmed that such an approach also increases the dependability of the water supply, hence minimizing the likelihood of drought. This result provides an answer to our first research question: "Can the suggested approach successfully alleviate drought in areas at high risk of drought?"

The adjusted variable neighborhood strategy adaptive search (A-VaNSAS) was developed to find an optimal solution to the model, and it provided solutions superior to those of the original search model. According to Table 7, the total number of water demand points supplied by the VaNSAS-solved model was 6.67% lower than that of the A-VaNSAS model. Other KPIs indicated that A-VaNSAS provides a superior solution to that of VaNSAS by 4.22–12.66%. It can therefore be concluded that the modified version of the improvement method selection formula is more effective than the original for the considered problem. Consequently, guidance from the current best solution can enhance the search capability of such an approach; this conclusion corresponds to the findings of Akararungruangkul, R. et al. [14], Sirirak, W. et al. [68], and Ketsripongsa, U. et al. [69]. The current solution is directed by the global best solution, and the solution quality of the modified version of the proposed methods (e.g., differential evolution algorithm) presented improved search performance. This provides a response to our second study question: "Can A-VaNSAS improve the quality of the original VaNSAS solution?"

The results demonstrated not only that A-VaNSAS outperforms the original VaNSAS, but also that it outperforms existing state-of-the-art heuristics (i.e., GA and DE). A-VaNSAS employs a range of improvement strategies that allow it to escape from local optima, potentially

making it more search-intensive in certain search spaces compared with the aforementioned methods. This concept has been reinforced by Thongkham, M. and Kaewman, S. [70], and Kaewman, S. and Akararungruangkul, R. [71]; by adding additional local search strategies to the proposed methods, it is possible to increase the quality of the search and find a better solution than methods that utilize fewer local search techniques while maintaining the same computational times. This informed us that in order to establish effective improvement procedures, the developed model must integrate a good and effective local search or improvement approach. The guidance of a good solution also plays an essential role in enhancing the solution quality of a method that can lead to a good search space.

Long-term drought risk management is difficult to achieve since, in the real world, a variety of unknown factors can prohibit community reservoirs from alleviating drought. Integrated assessment models that study uncertain future conditions and potential policy interventions can improve strategic decision-making in long-term drought risk management, as has been indicated by Mens, M. et al. [72]. For long-term drought management, one may integrate a national hydrological model, a national one-dimensional hydrodynamic model, a regional one-dimensional hydrodynamic model, or a national surface water temperature model, among others. These models must meet the various and frequently contradicting requirements of policymakers, model developers, and other stakeholders. A community reservoir model cannot be a complete success if it lacks other ties to the community, government connections, and the support of local politicians and policymakers. Serena, H. et al. [73]; Loucks, D.P. and van Beek, E. [74]; and Haasnoot, M. et al. [75] have all supported the notion of developing a model that includes several stakeholders in addition to the single stakeholder given in the model. The proposed model can be interpreted as the operational model, whereas the model described in the aforementioned articles is developed at the strategic level of planning, which may therefore produce a different model. The development of planning models at the strategic and operational levels requires the use of different data and the consideration of different objectives [76]. The strategic level of drought risk management may involve devising a method to solve a problem over a larger area or use long-term data that can incorporate a forecasting model, whereas the operational level of management is concerned with the proper placement of community reservoirs and their network to meet the demands of a smaller area, as demonstrated by the proposed model.

## 5. Conclusions and Future Research Opportunities

CR–WSND presents a challenge wherein a water supply network to service other settlements without a community reservoir (CR) must be developed. In this paper, this problem was represented using mixed-integer programming (MIP) and then solved using a modified version of the variable neighborhood approach adaptive search method (A-VaNSAS). A new black box selection formula was developed to replace the previous formula, as well as new A-VaNSAS black box improvement approaches. DE and GA were used for comparison, demonstrating the superiority of the proposed technique with respect to well-known heuristics.

All the tested methods were used to tackle a real-world problem involving 218 subdistricts in Khong Chiam, Thailand. Three different sizes of CR were considered for use in drought-prone areas. When a CR is established in a particular site, a water supply network must be built to fulfill the needs of the local community. The computational output included two options: (1) no budget constraints and (2) budget constraints. A-VaNSAS costs 7.78%, 14.19%, and 12.62% less than GA, DE, and VaNSAS when there was no limitation on budget, respectively, and all the demand nodes were supplied by at least one CR. Therefore, drought risk was eliminated as all the demand nodes were satisfied in this case. The investment value was about THB 18,764,760, which makes this approach impossible to carry out all at once, considering relevant budgets. The second experiment was executed with a lower budget, and new key performance indicators (KPIs) were used to track how the system evolved since not all the demand nodes were fulfilled, including

the number of locations in the water supply network (WSN); the number of locations with community reservoirs constructed; (3) the full capacity of the constructed reservoirs; (4) the total area covered by the WSN; (5) the total construction cost; and (6) the total drought risk mitigation incentive.

The results of the computation revealed that GA, DE, and VaNSAS produced lower values for all the KPIs than A-VaNSAS did. The WSN, GA, DE, and VaNSAS integrated 7.87–11.24% fewer demand nodes than A-VaNSAS did. The total quantity of water in the A-VaNSAS system was higher than that with other approaches, with a maximum increase of 9.38%. In terms of overall drought risk mitigation incentive maximization, A-VaNSAS outperformed all the other approaches, meaning that it could identify ideal locations for the community reservoirs, thus minimizing the risk of drought. Thus, it was concluded that A-VaNSAS may be used to plan the community water supply networks in the target area. Furthermore, A-VaNSAS can generate great results that are superior to those supplied by existing well-known heuristics such as GA, DE, and VaNSAS even when constrained by a low budget.

Extensions to this research may be undertaken in a variety of ways in the future. For example, (1) the subsidiary cost in this study was combined into a single objective function, allowing the model to be transformed into a multi-objective model in future research and (2) the study area can be narrowed to the micro-level (i.e., from sub-districts to villages), making it more practical for local government organizations to implement.

**Author Contributions:** Conceptualization, R.S. and R.P.; methodology, N.N. and T.S.; software, S.K. and W.S.; validation, C.T., R.S. and R.P.; formal analysis, C.T. and T.S.; investigation, S.K. and N.N.; resources, W.S. and N.N.; data curation, R.S.; writing—original draft preparation, N.N. and R.P.; writing—review and editing, C.T. and W.S.; visualization, S.K. and T.S.; supervision, R.P.; project administration, R.S. All authors have read and agreed to the published version of the manuscript.

**Funding:** This research received no external funding.

**Data Availability Statement:** Some or all data, models, or codes that support the findings of this study are available from the corresponding author upon reasonable request.

**Conflicts of Interest:** The authors declare no conflict of interest.

## Appendix A. Example of the Vector Used in the Proposed Method

| Node Track | 1 | 2 | 3 | 4 | 5 | 6 | 7 | 8 | 9 | 10 |
|---|---|---|---|---|---|---|---|---|---|---|
| 1 | 0.67 | 0.43 | 0.03 | 0.10 | 0.69 | 0.72 | 0.54 | 0.56 | 0.58 | 0.68 |
| 2 | 0.73 | 0.17 | 0.43 | 0.90 | 0.76 | 0.12 | 0.06 | 0.46 | 0.87 | 0.30 |
| 3 | 0.90 | 0.55 | 0.92 | 0.67 | 0.43 | 0.10 | 0.35 | 0.94 | 0.50 | 0.60 |
| 4 | 0.09 | 0.22 | 0.50 | 0.89 | 0.48 | 0.60 | 0.86 | 0.92 | 0.69 | 0.58 |
| 5 | 0.26 | 0.63 | 0.79 | 0.90 | 0.89 | 0.37 | 0.53 | 0.50 | 0.12 | 0.52 |

## Appendix B. Details of the Candidate Nodes

| Node no. | Water Requirements (for the Demand Node; Cubic Meters) | Drought Risk | Height above Sea Level (Meters) |
|---|---|---|---|
| 1 | 757.7 | 0.85 | 144 |
| 2 | 826.7 | 0.85 | 184 |
| 3 | 928.1 | 0.85 | 149 |
| 4 | 582.4 | 0.85 | 190 |
| 5 | 849 | 0.85 | 183 |
| 6 | 916.5 | 0.65 | 164 |
| 7 | 770.2 | 0.65 | 164 |
| 8 | 641.7 | 0.65 | 148 |
| 9 | 998.9 | 0.65 | 164 |
| 10 | 806.3 | 0.65 | 135 |

## Appendix C. Details of the Water Reservoir

| Type of Water Reservoir | Criteria Probability for Selection | Capacity (Cubic Meters) | Distance Limitation (Kilometer Limitation (Kilometers)) | Construction Cost (Baht) |
|---|---|---|---|---|
| SC | 0–0.33 | 1260 | 30 | 43,470 |
| LC | 0.34–0.66 | 3780 | 80 | 121,440 |
| AW | 0.67–1.00 | 4520 | 100 | 255,000 |

## Appendix D. Example of the K Transition Method

| Node Track # | 1 | 2 | 3 | 4 | 5 |
|---|---|---|---|---|---|
| Target track | 0.23 | 0.44 | 0.39 | 0.18 | 0.92 |
| Candidate track | 0.64 | 0.73 | 0.07 | 0.27 | 0.57 |
| New selected track | 0.23 | 0.73 | 0.39 | 0.27 | 0.92 |

## Appendix E. Example of the K Cyclic Move Method

| Node Track | 1 | 2 | 3 | 4 | 5 | 6 | 7 | 8 | 9 | 10 |
|---|---|---|---|---|---|---|---|---|---|---|
| Initial | 0.67 | 0.43 | 0.03 | 0.10 | 0.69 | 0.72 | 0.54 | 0.56 | 0.58 | 0.68 |
| After KCM #1 | 0.67 | 0.10 | 0.03 | 0.54 | 0.69 | 0.72 | 0.58 | 0.56 | 0.43 | 0.68 |
| After KCM #2 | 0.67 | 0.54 | 0.03 | 0.58 | 0.69 | 0.72 | 0.43 | 0.56 | 0.10 | 0.68 |
| After KCM #3 | 0.67 | 0.58 | 0.03 | 0.43 | 0.69 | 0.72 | 0.10 | 0.56 | 0.54 | 0.68 |

## Appendix F. Results of the GBT

| Elements Track # | 1 | 2 | 3 | 4 | 5 |
|---|---|---|---|---|---|
| Target track | 0.45 | 0.12 | 0.27 | 0.45 | 0.19 |
| Best track | 0.59 | 0.02 | 0.57 | 0.88 | 0.60 |
| New selected track | 0.45 | 0.02 | 0.57 | 0.45 | 0.19 |

## Appendix G. Pseudocode of A-VaNSAS

**Algorithm A1.** Adjusted variable neighborhood strategy adaptive search (A-VaNSAS)

**Input:** Number of tracks (NT), number of nodes (D), scaling factor (F), improvement factor (K), and number of improvement/black box (NBB)
**Output:** Best_Track_Solution
    Begin
        Population = Initialize Population (NT, D)
        IBPop = Initialize Information BB (NBB)
        Encode Population to WP
    **While** *the stopping criterion is not met* **carry out**
    **For** *i = 1: NT*
    *//selected improvement box by*
        *RouletteWheelSelection*
            *selected_BB = RouletteWheelSelection(IBPop)*
                *If (selected_BB = 1) Then new_u = SWAP(u) Perform SWAP*
                    *Else if (selected_BB = 2) new_u = 2-Opt (u) Perform 2-Opt*
            *Else if (selected_BB = 3) new_u = K-Transition (u) Perform K-Transition*
                *Else if (selected_BB = 4) new_u = K-Cyclic (u) Perform K-Cyclic*
        *If (CostFunction(new_u) ≤ CostFunction($V_i$)) Then $V_i$ = New_u*
        *//Loop update heuristic information of Black box*
    **For** *j = 1:NBB*
    *BBPopi = α\*(ABBi) + (1 − α)\*(GBBi) + β\*(NBBi)*
        ***End For Loop****//end update heuristics information*
  **End For Loop i**
    End While Loop
      Return Best_Track_Solution
        End

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
