# Peer review of "Community Agricultural Reservoir Construction and Water Supply Network Design in Ubon Ratchathani, Thailand, Using Adjusted Variable Neighborhood Strategy Adaptive Search"

_water, doi:10.3390/w15030591_

Round 1
Reviewer 1 Report (Previous Reviewer 2)
In this submission, the previous version of the manuscript was improved. The water tank sizing methodology is defined in order to solve the actual problems of the insufficient water amount required for the water supply.
The literature review shows the actual state of the art of the appropriate and relevant knowledge about water sizing related to the analyzed problem.
Authors should enclose the diagrams of the water pumping and outlet regime, which is the lack of paper.
Also, the authors should describe the pressure conditions in the water supply system.
Author Response
We would like to thank you the reviewer for your insightful feedback. All comments were resolved as details below.
|
Comment |
Answer |
|
Authors should enclose the diagrams of the water pumping and outlet regime, which is the lack of paper. |
We have added the diagram in Figure 4 (line 562). Additional information regards to the diagram is explained in line 563-587. |
|
Also, the authors should describe the pressure conditions in the water supply system. |
The water pressure is described on line 563 to 587. |
Reviewer 2 Report (New Reviewer)
The paper proposed the use of A-VaNSAS to a real-world cenario to improve water supply network desgin. It is interesting and informative. The major problem with the manuscript is presentation, there are excessive tables and figures in the mansucript, which is diffcult to follow. The authors should put the algorithm and less important figures and tables in the supporting information or in more combined mode to promote the flow of the manuscript, making it easier for the readers to follow.
Author Response
We would like to thank you for your insightful feedback. We have tried our best to resolve as details in the table below.
|
Comment |
Answer |
|
there are excessive tables and figures in the mansucript, which is diffcult to follow. The authors should put the algorithm and less important figures and tables in the supporting information or in more combined mode to promote the flow of the manuscript, making it easier for the readers to follow. |
In order to allow the manuscript to flow easily for the reader, we remove six tables and 1 Figure, Tables 3 through 9 and Figure 6, as an appendix A to G. Thank you very much for the valuable comment. |
Reviewer 3 Report (New Reviewer)
The work is an interesting study . The data preparation done by the authors are good . The paper is structured and written with moderate English standard level . I suggest the authors to use standard English to improve the quality of the content. I also suggest to cite more relevant recent literature. Enhance the quality of the results also.
Author Response
Thank you for your insightful feedback. All comments were resolved as described in the table below.
|
Comment |
Answer |
|
I suggest the authors to use standard English to improve the quality of the content. |
We sent the article to MDPI English service to edit again. The certification is attached. |
|
I also suggest to cite more relevant recent literature |
We added section 2.4 to reflect latest reservoir development and water supply network design-related studies. Six publications scheduled for release in 2023 have been reviewed and presented. (line 315-348) |
|
Enhance the quality of the results also. |
Figures 10 and 11 have been added to the most recent version. It concludes the advantages of utilizing the proposed model A-VaNSAS in comparison to the existing approach and demonstrates that A-VaNSAS outperformed all other ways. In addition, it is demonstrated that the proposed model and approaches can lower the drought risk area in the numerical result of the case study. (line 885-894) |

Round 2
Reviewer 1 Report (Previous Reviewer 2)
The authors have improved the paper in accordance with my comments. Everything is fine.
This manuscript is a resubmission of an earlier submission. The following is a list of the peer review reports and author responses from that submission.
Round 1
Reviewer 1 Report
Please review English writing problems.
Reviewer 2 Report
The paper deals with the interesting and very actual water supply problem for irrigation purposes. The analysis is provided on a real case study. A mathematical description of the algorithm in the methodology is done correctly.
The manuscript has some shortcomings, leading to my decision, which is a major revision. Here are the arguments for my decision.
-authors are analyzing irrigation for the open areas. I cannot find any calculation or including of the precipitation. Thailand's weather conditions are well known for their heavy rain. Such includes a great intensity and variability of the precipitation amounts.
-literature review is superficial. The authors mention examples from many countries, but none of them could be comparable to Thailand.
-how is the variability of the inflow and outflow into/out of the reservoir solved? Are water tanks sized on a daily base? Please elaborate on this.
-figures are blurry, especially the maps. Please correct this.
-authors should enclose maps of Thailand with an analyzed location.
-authors should avoid writing in the first face. For example ''we have'' should be ''it has been'' or a similar (neutral) sentence composition.
Reviewer 3 Report
This manuscript presents the proposal of A-VaNSAS to solve the problem of a water supply network to service other community settlements without a reservoir. In general, this research is largely a practical work to solve problems in Ubon Ratchathani, Thailand. The theoretical gap is not clear, and the explanation structure goes back and forth, which does not help to present linearity and clarity of scientific ideas to the audience/readers. If the authors intend to publish this manuscript as a scientific article, the following comments intend to enable the authors to disseminate their work at the highest possible quality.
I. LANGUAGE.
The language used in this manuscript is unnatural. There are unnecessarily long sentences than should be written as consecutive sentences for clarity. Besides, past, present, and perfect tenses must be checked to ensure the correct placements/usages throughout the text.
II. ABSTRACT.
Please state non-abbreviated terms correctly before being abbreviated. For example, what is GMS? In the case of the Agricultural Community’s Reservoir establishing and Water Supply Network Design, should it be ACR-WSND? What the word "establishing" is supposed to refer to in the original term? Other abbreviations that require further clarity are VaNSAS and A-VaNSAS.
Aside from that, please refer to the specific part of the Mekong Basin this study is focusing on (Lines 20-22). It would help avoid a misleading sense of the geographical boundary of this study.
III. INTRODUCTION.
This research does not address a clear research gap. The study merely considers practical situations in Ubon Ratchathani, Thailand, with no theoretical gap found in the literature. Consequently, this study is a mere exercise of a couple of algorithms and their comparison.
There is no theoretical necessity for why, for example, VaNSAS should be the algorithm in question. VaNSAS is never mentioned in the Introduction but suddenly became the approach in question at the heart of the study. There is also no theoretical necessity for why VaNSAS must be extended into A-VaNSAS.
Again, without a clear research gap (not practical gaps/problems in the observed location), the extension of VaNSAS into A-VaNSAS, including the comparison to DE and GA established in this study, does not have clear scientific necessities based on the literature.
At the end of this "INTRODUCTION" section, the authors should add a couple of research questions. These questions would ensure research processes and results consistent with the research aim. The authors may refer to this article to build appropriate research questions for this research ⇒ DOI: 10.1177/1350508410372151
As the consequences of the concerns above, the mentions of the Ubon Ratchathani and Mekong case should not appear in the "INTRODUCTION" section. They should appear as part of the "METHODOLOGY" and "MODEL TESTING" sections (see relevant comments below).
IV. LITERATURE REVIEW.
Currently, the "LITERATURE REVIEW" section is a mix of methodological explanations and a brief review of approaches/algorithms that would be considered in this study. There is no rigorous review on the relationships between "drought management" and "relevant algorithms" (not as separated explanations), and why VaNSAS eventually requires an "adjustment" (hence A-VaNSAS). Without a rigorous literature review on these matters, the "LITERATURE REVIEW" section is merely a review of theoretical foundations.
At the end of the "LITERATURE REVIEW" section, the authors must present the conceptual framework of adjusting VaNSAS into something else. After the "why" questions (why VaNSAS and why it should be adjusted) are answered in the early part of this section (as suggested above), the "what" question (what should be adjusted in VaNSAS) must be answered by proposing a conceptual framework of the A-VaNSAS. Without the conceptual framework, the "LITERATURE REVIEW" does not produce any meaningful contribution to the entire manuscript.
V. METHODOLOGY.
This study can be distinguished into two parts: methodological level (the proposal/model building of A-VaNSAS) and practical level (the model testing of A-VaNSAS). Therefore, this manuscript requires a dedicated "METHODOLOGY" section to explain how the gap finding (Introduction), conceptual model (Literature Review), model building (A-VaNSAS), and model testing (the case of Ubon Ratchathani) are arranged as the stages/steps of the entire study (research stages).
In the "METHODOLOGY" section, the authors must present briefly what each stage is, what it serves, what the purposes are, what the approaches used in the stage, and what the stage intends to produce. The explanations would produce a "Research Design," which would guide readers to understand how the authors develop the research itself. It would also help prove the repeatability of the research, making other studies able to repeat this study in different locations/cases.
For an easier explanation, consider making a brief explanation of each research stage as one subsection of the "METHODOLOGY" section. Also, consider including lines 154-186 (from "LITERATURE REVIEW"), the general explanations of the Mekong watershed and the Ubon Ratchathani case (from "INTRODUCTION") as part of the "Model Testing" subsection (part of the "METHODOLOGY" section).
VI. MODEL BUILDING.
The mathematical model for CR-SWSND (lines 342-456) must appear before line 274 (which should be the beginning of the "MODEL TESTING" section). The presentation of the model would stand as the "MODEL BUILDING" section to answer the "how" question (how the VaNSAS is adjusted).
VII. MODEL TESTING.
In this section, the authors should explain how the proposed A-VaNSAS model was applied/tested in the Ubon Ratchathani case, including what data were used for which specific variables in the proposed A-VaNSAS model. Consider making lines 274-340 the beginning of this section, followed by lines 457-612. Please do not forget to add bridging paragraph(s) to connect those currently-separated parts of the manuscript.
VIII. DISCUSSION.
This article lacks a "DISCUSSION" section to prove the theoretical contributions of this research to the scientific literature. After the "RESULTS" section (lines 613-878) presents the results of the "MODEL TESTING" (lines 274-340 and 457-612), this "DISCUSSION" section should present extensive comparisons between the findings of this study and the findings of relevant studies.
The comparisons must include how each finding is similar (confirmatory findings) or dissimilar (counterintuitive findings) to the results of published studies. That way, the authors can argue how this study stands among other literature within the body of knowledge. Despite having different topics, the authors may learn about some "active" discussions in these sample articles ⇒ DOIs 10.3390/su14084562; 10.3390/admsci12020048; 10.3390/su14031865
The "Discussion" section should be structurally divided into several subsections according to the research questions. Each subsection should refer to one specific research question. It would clarify which findings answer which research question, convincing readers that this study completely fulfils its aim and every objective.